# Splice-dependent trans-synaptic PTPδ–IL1RAPL1 interaction regulates synapse formation and non-REM sleep

Haram Park[1,†], Yeonsoo Choi[1,†], Hwajin Jung[1,†], Seoyeong Kim[2,†], Suho Lee[1], Hyemin Han[3], Hanseul Kweon[2], Suwon Kang[2], Woong Seob Sim[2], Frank Koopmans[4,5] (iD), Esther Yang[6], Hyun Kim[6], August B Smit[5], Yong Chul Bae[3] & Eunjoon Kim[1,2,*] (iD)

## Abstract

Alternative splicing regulates trans-synaptic adhesions and synapse development, but supporting *in vivo* evidence is limited. PTPδ, a receptor tyrosine phosphatase adhering to multiple synaptic adhesion molecules, is associated with various neuropsychiatric disorders; however, its *in vivo* functions remain unclear. Here, we show that PTPδ is mainly present at excitatory presynaptic sites by endogenous PTPδ tagging. Global PTPδ deletion in mice leads to input-specific decreases in excitatory synapse development and strength. This involves tyrosine dephosphorylation and synaptic loss of IL1RAPL1, a postsynaptic partner of PTPδ requiring the PTPδ-meA splice insert for binding. Importantly, PTPδ-mutant mice lacking the PTPδ-meA insert, and thus lacking the PTPδ interaction with IL1RAPL1 but not other postsynaptic partners, recapitulate biochemical and synaptic phenotypes of global PTPδ-mutant mice. Behaviorally, both global and meA-specific PTPδ-mutant mice display abnormal sleep behavior and non-REM rhythms. Therefore, alternative splicing in PTPδ regulates excitatory synapse development and sleep by modulating a specific trans-synaptic adhesion.

**Keywords** alternative splicing; receptor tyrosine phosphatase; sleep behavior and rhythm; synapse development; synaptic adhesion
**Subject Category** Neuroscience
**The EMBO Journal (2020) 39: e104150**

## Introduction

Synaptic cell adhesion molecules (CAMs) mediate trans-synaptic adhesions to regulate synapse development and function (Shen & Scheiffele, 2010; Siddiqui & Craig, 2011; de Wit & Ghosh, 2016; Südhof, 2017, 2018; Yuzaki, 2018; Kurshan & Shen, 2019). These trans-synaptic adhesions are often themselves regulated by small splice inserts present on the extracellular regions of synaptic adhesion molecules. Accordingly, alternative splicing has been shown to regulate aspects of synapse development and function, including the specificity and strength of trans-synaptic adhesions, excitatory or inhibitory synaptic localization of adhesion molecules, and trans-synaptic control of the responses of postsynaptic receptors, such as NMDA receptors (NMDARs) and AMPA receptors (AMPARs) (Boucard *et al*, 2005; Chih *et al*, 2006; Graf *et al*, 2006; Aoto *et al*, 2013; Dai *et al*, 2019).

Splicing-dependent regulation of synaptic adhesions has been extensively studied for neurexins (Schreiner *et al*, 2014; Treutlein *et al*, 2014). Neuronal activity regulates alternative splicing of neurexins through dedicated splicing program (Iijima *et al*, 2011; Shapiro-Reznik *et al*, 2012; Ehrmann *et al*, 2013; Traunmuller *et al*, 2016; Ding *et al*, 2017). The splice site 4 (SS4) insert on neurexins promotes or inhibits the binding of neurexins to postsynaptic partners, including neuroligins, cerebellins, LRRTMs, and latrophilins (Sugita *et al*, 2001; Boucard *et al*, 2005, 2012; Chih *et al*, 2006; Graf *et al*, 2006; Ko *et al*, 2009; Siddiqui *et al*, 2010; Uemura *et al*, 2010). Constitutive inclusion of the SS4 insert in presynaptic neurexin-3 decreases postsynaptic levels of AMPARs, but not NMDARs, in mice (Aoto *et al*, 2013). The SS4 insert on different neurexins (neurexin-1/2/3) distinctly regulates postsynaptic AMPAR- and NMDAR-mediated responses (Dai *et al*, 2019). Conversely, splice inserts on neuroligins regulate contacts of neuroligins with glutamatergic and GABAergic nerve terminals (Chih *et al*, 2006) as well as the binding preference of neuroligin-1 for α- or β-neurexins (Boucard *et al*, 2005).

The LAR (leukocyte antigen-related) family of receptor protein tyrosine phosphatases (LAR-RPTPs), comprising three known

1 Center for Synaptic Brain Dysfunctions, Institute for Basic Science (IBS), Daejeon, Korea
2 Department of Biological Sciences, Korea Advanced Institute for Science and Technology (KAIST), Daejeon, Korea
3 Department of Anatomy and Neurobiology, School of Dentistry, Kyungpook National University, Daegu, Korea
4 Department of Functional Genomics, CNCR, VU University and UMC Amsterdam, Amsterdam, The Netherlands
5 Department of Molecular and Cellular Neurobiology, CNCR, VU University and UMC Amsterdam, Amsterdam, The Netherlands
6 Department of Anatomy and Division of Brain Korea 21, Biomedical Science, College of Medicine, Korea University, Seoul, Korea
  *Corresponding author. Tel: +82 42 350 2633; E-mail: kime@kaist.ac.kr
  †These authors contributed equally to this work

members, *Ptprf*/LAR, *Ptprs*/PTPσ, and *Ptprd*/PTPδ, are synaptic adhesion molecules that possess tyrosine phosphatase activity in their cytoplasmic region (Takahashi & Craig, 2013; Um & Ko, 2013). LAR-RPTPs have been suggested to regulate synapse development and function by acting as presynaptic organizers that trans-synaptically interact with multiple postsynaptic CAMs, including NGL-3 (netrin-G ligand-3), TrkC, Slitrks, IL1RAPL1 (interleukin 1 receptor accessory protein like 1), IL-1RAcP (interleukin 1 receptor accessory protein), and SALM3/5 (cell adhesion-like molecule 3/5, also known as LRFN4/5 for leucine-rich repeat and fibronectin type III domain) (Woo *et al*, 2009; Kwon *et al*, 2010; Takahashi *et al*, 2011, 2012; Valnegri *et al*, 2011; Yoshida *et al*, 2011, 2012; Yim *et al*, 2013; Li *et al*, 2015; Choi *et al*, 2016). These interactions are frequently regulated by splice inserts located on the extracellular regions of LAR-RPTPs; related molecular details have been clarified by biochemical, cell biological, and X-ray crystallographic studies (Coles *et al*, 2014; Um *et al*, 2014; Yamagata *et al*, 2015a; Won *et al*, 2017; Goto-Ito *et al*, 2018; Karki *et al*, 2018; Lin *et al*, 2018). For instance, PTPδ requires the miniexon A (meA) splice insert to interact with IL1RAPL1 (Yoshida *et al*, 2011; Yamagata *et al*, 2015b), a postsynaptic CAM that critically regulates synaptic and neuronal functions (Montani *et al*, 2019), whereas PTPδ requires the miniexon B (meB) to interact with Slitrks, SALM3/5, and IL-1RAcP (Takahashi *et al*, 2012; Yoshida *et al*, 2012; Yim *et al*, 2013; Um *et al*, 2014; Li *et al*, 2015; Yamagata *et al*, 2015a,b; Choi *et al*, 2016).

Although these observations provide important clues regarding the functions of LAR-RPTPs, it remains largely unclear whether LAR-RPTPs, under physiological conditions, are expressed in particular brain regions and cell types, distribute to pre- or postsynaptic sites, are present at excitatory or inhibitory synaptic sites, regulate synapse development and function, and/or employ alternative splicing for synapse regulation. Beyond these fundamental aspects, LAR-RPTPs have been implicated in diverse brain disorders. In particular, *PTPRD* has been extensively associated with a large number of brain disorders (Uhl & Martinez, 2019), including attention-deficit hyperactivity disorder (ADHD) (Anney *et al*, 2008; Elia *et al*, 2010; Distel *et al*, 2011; Jarick *et al*, 2014), restless leg syndrome (SNP study) (Schormair *et al*, 2008; Yang *et al*, 2011; Kim *et al*, 2013; Moore *et al*, 2014), addiction (Drgonova *et al*, 2015; Uhl *et al*, 2018), bipolar disorder (Malhotra *et al*, 2011), obsessive–compulsive disorder (Mattheisen *et al*, 2015), and intellectual disability (Choucair *et al*, 2015). Previous studies on *Ptprd*-mutant mice have revealed impairments in synaptic plasticity, learning, and memory (Uetani *et al*, 2000; Drgonova *et al*, 2015); dendritic arborization in cortical neurons (Nakamura *et al*, 2017); and axon targeting in motor neurons (Uetani *et al*, 2006), as well as enhanced cocaine preference and suppressed sleep-like behaviors (Drgonova *et al*, 2015). However, *in vivo* impacts of alternative splicing on PTPδ-dependent trans-synaptic interactions and functions have not been explored.

Here, we generated a knock-in mouse line carrying the endogenous PTPδ protein tagged with tdTomato and found that PTPδ signals are detectable in widespread brain regions and at excitatory presynaptic sites in the hippocampus. Another knockout mouse line globally lacking PTPδ (PTPδ-KO) showed an input-specific decrease in excitatory synapse density and strength in distal dendrites of hippocampal CA1 neurons. These changes were associated with robust decreases in tyrosine phosphorylation and excitatory synaptic localization of IL1RAPL1, which requires the PTPδ-meA splice insert for PTPδ binding. These synaptic and biochemical phenotypes were recapitulated by another knockout mouse line lacking the PTPδ-meA insert and thus PTPδ–IL1RAPL1 interactions. In addition, PTPδ-KO and meA-specific PTPδ-mutant mice displayed shared abnormalities in sleep behavior and rhythms. These data provide *in vivo* support for the role of alternative splicing in the regulation of excitatory synapse development and sleep behavior and rhythms.

# Results

## Widespread expression of PTPδ proteins in the mouse brain

To better understand *in vivo* functions of PTPδ, we fluorescently tagged endogenous PTPδ protein by fusing tdTomato to the C-terminus of PTPδ (PTPδ-tdTomato), generating a new PTPδ-reporter mouse line (Fig 1A). Western blot analysis of whole-brain lysates from these mice confirmed gene dosage-dependent expression of the PTPδ-tdTomato protein (~140 kDa) at the expense of the endogenous PTPδ protein (~85 kDa) (Fig 1B). The reduced levels of PTPδ-tdTomato proteins relative to endogenous PTPδ proteins, both of which represent the C-terminal fragment after proteolytic cleavage in the middle of the protein (Chagnon *et al*, 2004), could be attributable to the destabilization of the hybrid PTPδ-tdTomato protein. Alternatively, the tdTomato portion of the PTPδ-tdTomato protein may suppress the C-terminal PTPδ antibody to bind to its cognate antigen. Notably, the N-terminal PTPδ antibody detected similar amounts of the N-terminal fragment of PTPδ-tdTomato (after cleavage) (Fig 1B).

Confocal imaging of brain slices from PTPδ-reporter mice revealed strong PTPδ-tdTomato signals in various brain regions (Figs 1C and EV1A–C). The signals were particularly strong in cortical layer I, the corpus callosum, and the anterior commissure—all of which feature dense axonal tracts—as well as the stratum lacunosum moleculare (SLM) and dentate gyrus molecular (DG-MO) layer of the hippocampus, upper cortical layers, thalamus, and the reticulate nucleus of the thalamus (TRN). Modest PTPδ-tdTomato signals were observed in the olfactory bulb, hypothalamus, and deeper cortical layers. In line with the previous suggestion for presynaptic functions of PTPδ, PTPδ-tdTomato signals were stronger in brain regions enriched with axonal fibers such as the corpus callosum, as supported by immunoblot analysis of corpus callosum samples (~50% higher in average intensity compared with the whole-brain sample, $n = 4$ mice, $P = 0.0016$, paired *t*-test). These results suggest widespread distribution patterns of PTPδ in the mouse brain.

## PTPδ-tdTomato protein at glutamatergic axon terminals

It has been suggested that PTPδ acts as a presynaptic organizer that interacts with several postsynaptic adhesion molecules (NGL-3, Slitrks, IL1RAPL1, IL-1RAcP, and SALM3/5) (Takahashi & Craig, 2013; Um & Ko, 2013), but other reports have implicated PTPδ in postsynaptic or dendritic regulation (Dunah *et al*, 2005; Shishikura *et al*, 2016; Montani *et al*, 2017; Nakamura *et al*, 2017). In addition, whether PTPδ is present primarily at excitatory or inhibitory nerve terminals remains unclear, largely owing to the lack of specific and high-affinity antibodies. Although previous studies have implicated PTPδ in the regulation of both excitatory and inhibitory

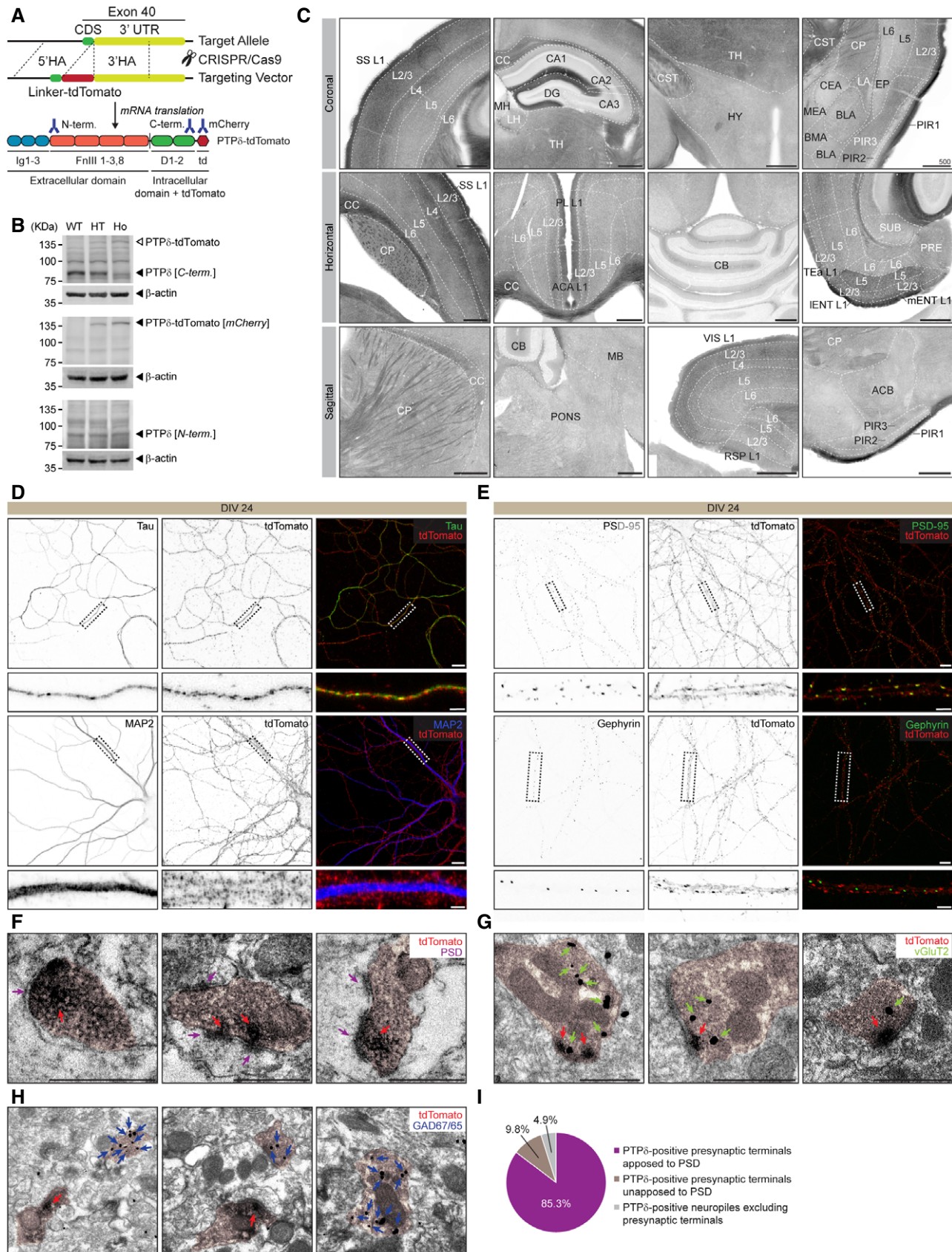

**Figure 1.**

**Figure 1.   PTPδ-tdTomato protein mainly localizes to presynaptic sites at excitatory, but not inhibitory, synapses in the mouse hippocampus.**

A   Transgenic strategy for generating PTPδ-tdTomato reporter mice used to visualize the distribution patterns of endogenous PTPδ protein. CRISPR/Cas9 was used to fuse tdTomato in frame to the C-terminus of the PTPδ protein, encoded by the last coding exon (exon 21) of the *Ptprd* gene. HA, homology arm; CDS, coding domain sequence; UTR, untranslated region; Ig, Immunoglobulin domain; FnIII, fibronectin 3-like domain; D, phosphatase domain; td, tdTomato.

B   Verification of PTPδ-tdTomato reporter mice by Western blot analysis of whole-brain lysates from heterozygote (HT) and homozygous (Ho) mice (P21) and PTPδ using mCherry antibodies. Note that levels of endogenous PTPδ protein (~85 kDa) are decreased, and that PTPδ-tdTomato protein (~140 kDa) is detectable only in reporter mice (HT and Ho). Note also that the 85-kDa band represents the transmembrane domain + cytoplasmic fragment of the full-length PTPδ protein that undergoes proteolytic cleavage at extracellular membrane-proximal sites, which can only be recognized by the PTPδ C-terminal antibody that targets the cytoplasmic region. The PTPδ-tdTomato protein (~140 kDa) contains the membrane domain + cytoplasmic fragment of PTPδ (~85 kDa) fused to tdTomato (55 kDa). The N-terminal antibody recognizes equal amounts of the C-terminal cleavage product derived from PTPδ and PTPδ-tdTomato proteins.

C   Representative coronal, horizontal, and sagittal sections from PTPδ-tdTomato mice. Enlarged windows are demarcated by black dotted lines (Fig EV1A–C). Red tdTomato images were converted to grayscale images for clarity. ACA, anterior cingulate area; ACB, nucleus accumbens; BLA, basolateral amygdala; BMA, basomedial amygdala; CA1, cornu ammonis area 1; CA2, cornu ammonis area 2; CA3, cornu ammonis area 3; CB, cerebellum; CC, corpus callosum; CEA, central amygdala; CP, caudate putamen; CST, corticospinal tract; DG, dentate gyrus; EP, endopiriform nucleus, HY, hypothalamus; L1, cortical layer 1; L2/3, cortical layer 2/3; LA, lateral amygdala; lENT, lateral entorhinal area; LH, lateral habenula; MB, midbrain; MEA, medial amygdala; mENT, medial entorhinal area; MH, medial habenula; PIR1, piriform molecular layer; PIR2, piriform pyramidal layer; PIR3, piriform polymorph layer; PL, prelimbic; PONS, pons; PRE, presubiculum; RSP, retrosplenial area; SS, somatosensory area; SUB, subiculum; TEa, temporal association area; TH, thalamus; VIS, visual area; Scale bar, 500 □m.

D   PTPδ-tdTomato protein is detected in tau-positive axonal compartments, but not in MAP2-positive dendritic compartments in cultured neurons (entorhinal cortex + hippocampus; 1:2 mixture; days *in vitro* or DIV 24) derived from *Ptprd*$^{-/-}$ mouse embryos. Note that the tdTomato signals envelop, but not overlap with, MAP2-delineated dendrites, although this does not necessarily suggest that tdTomato signals not in dendrites. Scale bars, 10 μm in main images and 2 μm in enlarged images.

E   Presynaptic PTPδ-tdTomato clusters colocalize more strongly with PSD-95 clusters (excitatory postsynaptic marker) relative to gephyrin (inhibitory postsynaptic marker) clusters in cultured neurons (entorhinal cortex + hippocampus; DIV24) derived from *Ptprd*$^{-/-}$ mouse embryos. See Fig EV1F for the quantification of the results. Scale bars, 10 μm in main images and 2 μm in enlarged images.

F–I   Ultrastructural localization of PTPδ-tdTomato protein (red arrows; DAB staining) at axon terminals juxtaposed to electron-dense postsynaptic densities (purple arrows; PSDs) and also at vGlut2-positive excitatory synaptic axon terminals (green arrows; immunogold staining) but not at GAD67/65-positive inhibitory synaptic axon terminals (blue arrows; immunogold staining) in the stratum lacunosum-moleculare (SLM) region of the CA1 hippocampal region (P21). Note that ~85% of PTPδ-tdTomato signals are present in excitatory presynaptic terminals apposed to PSDs, ~10% at presynaptic terminals not apposed to PSDs, and ~5% in neuropils other than presynaptic terminals. Axon terminals are indicated by pink shades. Scale bar, 500 nm (*n* = 4 areas from 2 mice [WT and KO]).

Source data are available online for this figure.

synaptic regulation, this inferred role is largely based on the nature of the postsynaptic partners of PTPδ studied (NGL-3, Slitrks, IL1RAPL1, IL-1RAcP, and SALM3/5) (Woo *et al*, 2009; Kwon *et al*, 2010; Valnegri *et al*, 2011; Yoshida *et al*, 2011, 2012; Takahashi *et al*, 2012; Yim *et al*, 2013; Li *et al*, 2015; Choi *et al*, 2016).

To more directly answer these questions, we employed cultured neurons (cortical + hippocampal) derived from PTPδ-tdTomato reporter mice. Immunocytochemical analyses of cultured PTPδ-tdTomato neurons revealed the presence of reporter proteins in both glutamatergic and GABAergic neurons, marked by vGluT1 and GAD67 expression, respectively (Fig EV1D). Intracellularly, PTPδ-tdTomato signals were prominently detected in tau-positive axons, but were largely absent in MAP2-positive dendrites, only largely enveloping the dendrites, at 24 days *in vitro* (DIV24) (Fig 1D), although the latter does not necessarily suggest that PTPδ-tdTomato signals are not present in dendrites because MAP2 signals are generally in the center region of dendrites. In contrast, there was no strongly polarized distribution in immature neurons (DIV6), where the reporter protein was detected in both dendrites and axons (Fig EV1E). These results suggest that PTPδ is expressed in both glutamatergic and GABAergic neurons, and is distributed to both axons and dendrites in immature neurons, but mainly to axons in mature neurons.

We next tested if PTPδ-tdTomato clusters are localized at excitatory or inhibitory presynaptic nerve terminals using cultured neurons, and found that PTPδ-tdTomato clusters were more strongly (~2.7-fold) in contact with PSD-95 relative to gephyrin clusters, marking excitatory and inhibitory synapses, respectively (Figs 1E and EV1F). Because neuronal cultures do not fully mimic the *in vivo* environment, we turned to electron microscopic (EM) analyses of the SLM region in brain slices. These experiments

showed strong PTPδ-tdTomato reporter signals in distal dendrites of CA1 hippocampal neurons (Fig 1C). Specifically, PTPδ-tdTomato diaminobenzidine (DAB) signals were detected mainly in neurotransmitter vesicle-rich axon terminals juxtaposed to postsynaptic densities (PSDs) (Figs 1F and EV1F). When combined with immunogold staining for glutamatergic (vGluT2) and GABAergic (GAD1/2) axon terminals, PTPδ-tdTomato signals were mainly (~85%) found in glutamatergic axon terminals (Figs 1G and H, and EV1G–I). PTPδ-tdTomato and endogenous PTPδ proteins were similarly distributed in subcellular brain fractions (Fig EV1J), suggesting that the addition of tdTomato to the end of PTPδ is less likely to alter the trafficking, synaptic localization, or protein interactions of PTPδ. These results suggest that PTPδ is mainly present at excitatory presynaptic sites in the hippocampal SLM region. However, these results do not exclude the possibility that PTPδ proteins are present at inhibitory synapses in addition to excitatory synapses in brain regions other than the hippocampal SLM.

### *Ptprd* deletion leads to input-specific decreases in excitatory synapse density and strength

Because PTPδ-tdTomato signals seemed to be enriched in specific laminar structures of the hippocampus (Fig 1C), we quantified the distribution of PTPδ-tdTomato signals in hippocampal layers. We found strong signals in the SLM region of the CA1 and the MO layer of the DG (Fig 2A), both of which receive glutamatergic inputs from the entorhinal cortex through the temporoammonic (TA) and perforant pathways. In parallel, we manually micro-dissected the SR (stratum radiatum) and SLM regions of wild-type (WT) CA1 slices and performed immunoblotting for PTPδ. These analyses indicated a stronger (~3-fold) enrichment of PTPδ in the SLM region relative to the SR (Fig 2B).

Given the clear enrichment of PTPδ in the CA1 SLM region, we next tested whether PTPδ is important for excitatory synaptic structure and transmission *in vivo*. To this end, we generated a new *Ptprd*-knockout mouse line (*Ptprd*−/− mice) carrying a global deletion of the *Ptprd* gene by removal of exon 13, which encodes the first Ig domain present in all known splice variants of the PTPδ protein (Mizuno *et al*, 1993; Pulido *et al*, 1995) (Fig 2C). This *Ptprd* deletion led to gene dosage-dependent decreases in PTPδ protein in immunoblot analyses of whole-brain lysates using PTPδ antibodies targeting N- and C-termini (Fig 2D).

Using these *Ptprd*−/− mice, we first measured spontaneous synaptic transmission at excitatory and inhibitory synapses in CA1 pyramidal neurons. However, we found no changes in miniature excitatory postsynaptic currents (mEPSCs), miniature inhibitory postsynaptic currents (mIPSCs), or spontaneous excitatory postsynaptic currents (sEPSCs) in *Ptprd*−/− CA1 neurons (Fig EV2A–D).

These results may indicate that (a) CA1 pyramidal cells receive postsynaptic currents from all three strata (of which the SR is the biggest contributor) and (b) signals from the SLM—the distal stratum in the CA1 most likely to be directly affected by PTPδ deletion—is subject to the cable effect, which severely attenuates the signal during it travels the dendrite (Dudman *et al*, 2007). Therefore, we measured input/output (I/O) ratio and paired-pulse facilitation (PPF) in each of the three strata. The SLM region showed a significant decrease (~20%) in the I/O ratio but not PPF, whereas no alterations in either parameter were observed in the SR or SO (Fig 2E–J). Interestingly, the slope of field excitatory postsynaptic potentials (fEPSPs), but not fiber volley, was reduced in the SLM. These results suggest that PTPδ deletion suppresses excitatory postsynaptic responses, but not axonal conduction or presynaptic release.

Structurally, an EM analysis of the *Ptprd*−/− SLM revealed a significant reduction (~40%) in PSD density, but not length, thickness, or perforation (a measure of maturation) (Fig 2K–O). In addition, the density of presynaptic nerve terminals was decreased (~28%) in the *Ptprd*−/− SLM, whereas the area of nerve terminals and the density of presynaptic vesicles were normal (Fig 2P–R), suggesting that presynaptic structures are substantially eliminated together with postsynaptic structures. These results suggest that the decreased I/O ratio in the SLM is attributable to a decrease in the number, but not function, of excitatory synapses. In contrast, the number and morphology of GABAergic synapses were normal in the *Ptprd*−/− SLM (Fig EV2E–H). Collectively, these results suggest that *Ptprd* deletion decreases excitatory synapse density and excitatory synaptic transmission in the CA1 SLM region in a lamina-specific manner.

### Altered tyrosine phosphorylation in multiple pre- and postsynaptic proteins in the *Ptprd*−/− brain

To explore the mechanisms underlying the input-specific decrease in excitatory synaptic density and transmission in the *Ptprd*−/− SLM region, but not the SR, we first performed comparative immunoblot analyses of total lysates of micro-dissected SLM and SR layers. We found, however, that levels of various synaptic proteins were largely normal in both the SLM region and SR of *Ptprd*−/− mice (Fig EV3A–F). Specifically, there were no changes in the levels of known postsynaptic binding partners of PTPδ (IL1RAPL1, Slitrk2/3, or NGL-3) or synaptic scaffolding/receptor/signaling proteins, including postsynaptic scaffolds (PSD-95 and SynGAP1), presynaptic scaffolds (Bassoon and liprin-α), glutamate receptor subunits (GluA1/2, GluN1, and GluN2A/B), and signaling molecules (CaMKII-α/β and p-Src). Notably, levels of PTPσ, a relative of PTPδ that is more strongly expressed in the SR relative to the SLM region (~3-fold), were increased ~2-fold in the SLM (Fig EV3D), likely representing a compensatory change. The lack of changes in the levels of major postsynaptic proteins in the SLM region of *Ptprd*−/− mice may be attributable to that total lysates, but not synapse-enriched preparations, were used for immunoblot analyses (see the results in Fig 4 below).

We next used an unbiased proteomic approach to investigate how *Ptprd* deletion decreases excitatory synapse density. Specifically, because the cytoplasmic region of PTPδ possesses tyrosine phosphatase activity, we examined changes in phosphotyrosine (pTyr) levels. As expected, these analyses indicated increased pTyr levels in multiple proteins (49 unique protein names/44 unique protein groups/59 unique peptides; absolute fold change > 1.5, $P < 0.05$) in *Ptprd*−/− whole-brain lysates. However, they also showed decreased pTyr levels in many other proteins (47 unique protein names/43 unique protein groups/49 unique peptides) (Fig 3A; Dataset EV), which likely represents indirect changes. The altered pTyr levels were observed at both known and novel amino acid residues in these proteins.

An analysis of proteins with altered pTyr levels (165 proteins; $P < 0.05$) using SynGO (synaptic Gene Ontology), a GO analysis that uses evidence-based and expert-curated GO terms based on experimental data on detailed synaptic localizations and functions (Koopmans *et al*, 2019), revealed that 57 proteins of the 167 proteins were annotated in SynGO, of which 45 had synapse-related functional annotations, including presynaptic (15 proteins), postsynaptic (7), and synapse organization (19) (Fig 3B).

Additional SynGO analysis focusing on the localization at specific sub-synaptic sites indicated that 25 proteins of the 49 synaptic proteins were presynaptic and 30 proteins were postsynaptic, including proteins that reside in both compartments (Fig 3C). In addition, these pre- and postsynaptic proteins displayed both up- and downregulated pTyr levels (Fig 3D and E). The presynaptic proteins with increased pTyr levels, which might be substrates of PTPδ, included well-known presynaptic proteins such as GAD67 (Lu *et al*, 2005), synapsin II (Thiel *et al*, 1990), PMCA1 (plasma membrane calcium-transporting ATPase 1) (Boyken *et al*, 2013), and EphA4 (Bouvier *et al*, 2008). Notably, the presynaptic proteins with decreased pTyr levels were often core components of the active zone (Gundelfinger & Fejtova, 2012; Sudhof, 2012; Gundelfinger *et al*, 2015), including Piccolo, Bassoon, Syt7 (synaptotagmin 7), Unc13A, caskin1 (CASK-interacting protein 1), and annexin A5. That LAR-RPTPs (PTPδ, PTPσ, and LAR) were also found in this group is likely attributable to the knockout of PTPδ protein.

The postsynaptic proteins with increased pTyr levels included serine/threonine kinases (CaMKIIα/β and JNK1/2/3) and receptor tyrosine kinases (TrkB, TrkC, EphA4). The postsynaptic proteins with decreased pTyr levels included IL1RAPL1, a postsynaptic binding partner of PTPδ important for excitatory synapse development (Pavlowsky *et al*, 2010a; Valnegri *et al*, 2011; Yoshida *et al*, 2011; Yasumura *et al*, 2014) that showed the greatest decrease in pTyr levels, as well as well-known PSD proteins, including PSD-95/93, Shank1/2/3, SynGAP1, GluN2B, and tyrosine kinases (Fyn, Lyn,

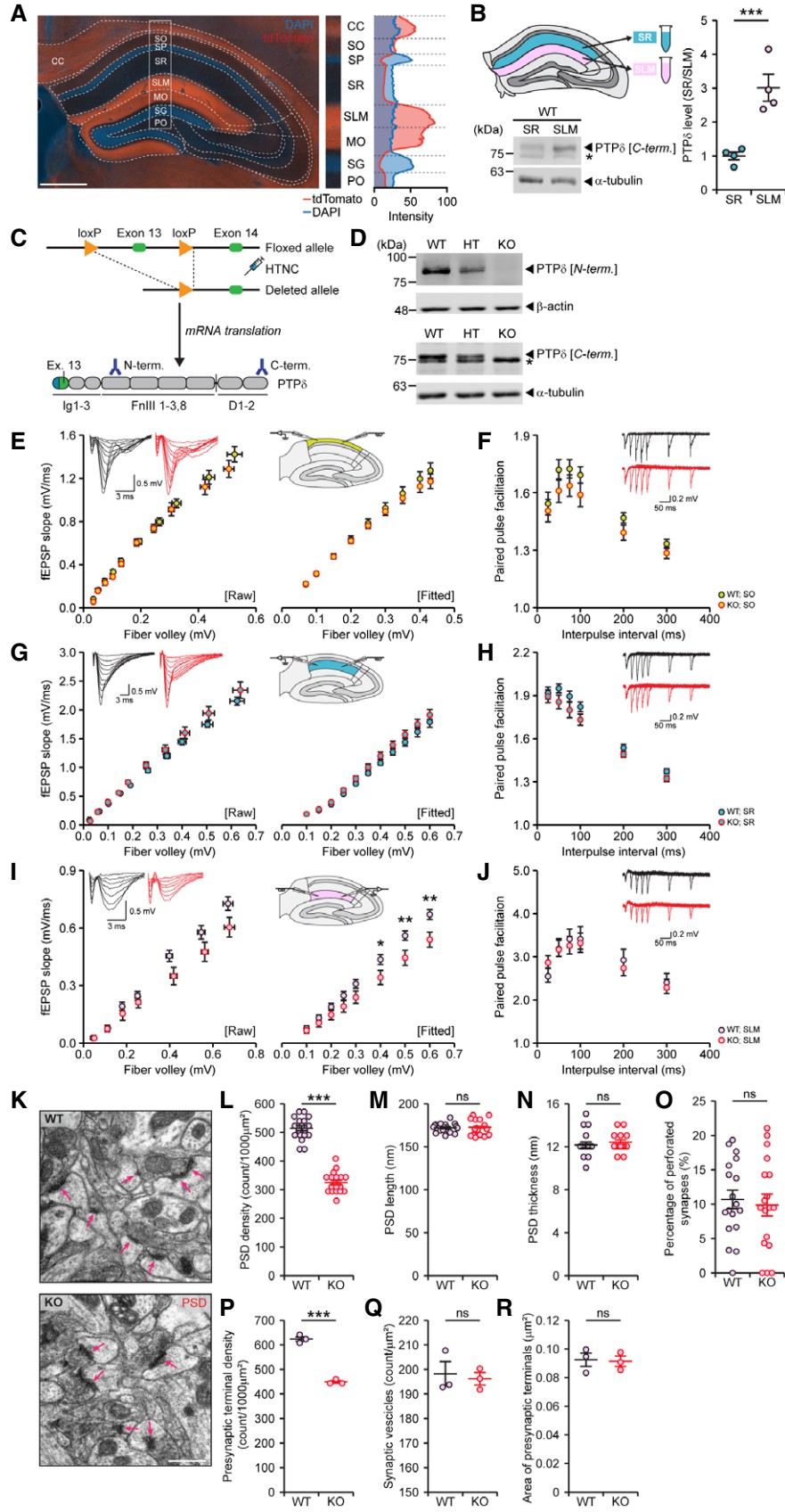

**Figure 2.**

◀ **Figure 2.** *Ptprd* deletion leads to input-specific suppression of excitatory synapse density and strength in the hippocampal SLM layer.

A  Strong PTPδ-tdTomato (red) signals in the SLM layer of the CA1 region and the MO layer of the DG in the hippocampus (2 months), revealed by quantitative analysis of PTPδ-tdTomato fluorescence across hippocampal lamina. DAPI was used for nuclear staining. CC, corpus callosum; SO, stratum oriens; SP, stratum pyramidale; SR, stratum radiatum; SLM, stratum lacunosum moleculare; MO, molecular layer; SG, granule cell layer; PO, polymorph layer. Scale bar 500 μm.

B  Biochemical enrichment of PTPδ protein in the hippocampal SLM layer relative to the SR layer. Micro-dissected SLM and SR lysates (2 months) were immunoblotted for PTPδ (C-terminal antibodies) and α-tubulin (control). The asterisk indicates a non-specific band. Analysis of immunoblots reveals significant enrichment of PTPδ in the SLM compared to the SR (n = 4 mice for SR and SLM, mean ± SEM, ***P < 0.001, Student's *t*-test; see Dataset EV2 for statistical details for these and all following results).

C  Schematic diagram for global *Ptprd* KO in mice, targeting exon 3. HTNC (His-tagged, TAT-fusion *Cre* with nuclear localization signal), carrying a membrane-permeable Cre recombinase, was used to remove exon 13 in fertilized eggs (see Methods for details). Sites on the PTPδ protein corresponding to epitopes recognized by anti-PTPδ antibodies (Ig1 domain [exon 13]) and N- and C-termini) are indicated.

D  Gene dose-dependent reductions in PTPδ protein levels in whole brains of *Ptprd*$^{+/-}$ (HT) and *Ptprd*$^{-/-}$ (KO) mice relative to *Ptprd*$^{+/+}$ (WT) mice (P21), shown by immunoblot analyses. Note that the C-terminal antibody also detects a non-specific band of similar size indicated by an asterisk, possibly either PTP( or LAR.

E–J  (E, G, and I) Suppressed basal excitatory synaptic transmission in the *Ptprd*$^{-/-}$ hippocampal SLM layer, but not SO or SR layer (2 months; male), as shown by the I/O ratio (n = 14 slices from 5 mice [WT-SO], 14, 5 [KO-SO], 11, 4 [WT-SR], 12, 5 [KO-SR], 14, 6 [WT-SLM], 12, 6 [KO-SLM], mean ± SEM, *P < 0.05; **P < 0.01, two-way repeated-measures/RM ANOVA, Holm-Sidak post-hoc multiple comparisons test). (F, H, and J) Normal presynaptic release in the *Ptprd*$^{-/-}$ hippocampal SLM, SO, and SR layers (P22–27; male), as shown by the PPF ratio (n = 13 slices from 5 mice [WT-SO], 13, 5 [KO-SO], 10,4 [WT-SR], 12, 5 [KO-SR], 15, 6 [WT-SLM], 14, 6 [KO-SLM], mean ± SEM, ns, not significant, two-way RM ANOVA).

K–O  Decreased PSD density, but not PSD length, thickness, or perforation (%), in the *Ptprd*$^{-/-}$ SLM (P21; male), revealed by EM. Arrows in panel K indicate PSDs (n = 18 images from 3 mice [WT and KO], ***P < 0.001, ns, not significant, Student's *t*-test for L, O-R, Mann-Whitney U test for M, N). Scale bar, 500 nm.

P–R  Decreased density of presynaptic nerve terminals but normal area of nerve terminals and normal density of presynaptic vesicles in the *Ptprd*$^{-/-}$ SLM (P21; male), revealed by EM (n = 3 mice [WT and KO], mean ± SEM, ***P < 0.001, ns, not significant, Student's *t*-test).

Source data are available online for this figure.

---

Yes, Src, and Pyk2 [(Giralt *et al*, 2017)]) (reviewed in (Sheng & Hoogenraad, 2007)).

These results collectively suggest that *Ptprd* deletion leads to widespread changes in pTyr levels in synaptic as well as non-synaptic proteins. Although some changes were expected, such as increased pTyr levels in presynaptic proteins, other changes were not (e.g., altered pTyr levels in postsynaptic proteins). Notably, PTPδ deletion also strongly downregulated pTyr levels of its postsynaptic partner, IL1RAPL1.

**PTPδ trans-synaptically regulates tyrosine phosphorylation and synaptic localization of IL1RAPL1 and excitatory synaptic strength**

To further explore how PTPδ deletion in presynaptic sites decreases excitatory synapse density in the hippocampus, we tested the possibility that IL1RAPL1 plays a role because it showed the strongest change—a ~15- to 20-fold decrease—in pTyr levels at two cytoplasmic residues (Y614 and Y648) (Fig 3A), far outstripping the protein with the second-highest change in pTyr status, PDHA1 (2- to 4-fold increase). In addition, IL1RAPL1 has been shown to important for normal excitatory synapse density in cultured neurons and the hippocampus (Pavlowsky *et al*, 2010a,b; Valnegri *et al*, 2011; Yoshida *et al*, 2011; Yasumura *et al*, 2014; Montani *et al*, 2017), although other postsynaptic partners of PTPδ, such as NGL-3, SALM3/5, and Slitrk2/3, have also been shown to regulate excitatory synapse density (Takahashi *et al*, 2012; Yim *et al*, 2013; Li *et al*, 2015; Choi *et al*, 2016; Lie *et al*, 2016).

We first tested whether IL1RAPL1 indeed displayed decreased pTyr levels in the whole brain of *Ptprd*$^{-/-}$ mice by immunoprecipitating IL1RAPL1 protein followed by immunoblotting for pTyr. This revealed strong tyrosine dephosphorylation of the IL1RAPL1 protein compared with that in the WT brain (Fig 4A). In addition, this change was associated with decreases in the distribution of IL1RAPL1 protein in the postsynaptic density (PSD) fraction and, to a lesser extent, in the synapse membrane fraction (SPM), although

IL1RAPL1 was normally enriched in the crude synaptosomal (P2) fraction (Fig 4B). In contrast, other PTPδ-binning postsynaptic adhesion molecules such as Slitrk3, IL1RAcP, and NGL-3 showed normal levels of enrichment in P2, SPM, and PSD fractions, comparable to those in WT samples (Fig EV3G). These results suggest that PTPδ is required for tight postsynaptic localization of IL1RAPL1 in the whole brain, including the hippocampus.

To further explore the possibility that PTPδ trans-synaptically regulates pTyr levels of IL1RAPL1, we generated another transgenic mouse line (*Ptprd-meA*$^{-/-}$) in which the PTPδ–IL1RAPL1 interaction was selectively abolished by deleting the meA splice insert (9 aa-long) of PTPδ, encoded by exons 15 and 16 (Fig 4C and D). The PTPδ-meA insert is critical for the PTPδ–IL1RAPL1 interaction but not interactions of PTPδ with other postsynaptic CAMs (NGL-3, Slitrks, IL1RAcP, SALM3/5, and IL-1RAcP) (Yoshida *et al*, 2011; Takahashi *et al*, 2012; Um *et al*, 2014; Li *et al*, 2015; Yamagata *et al*, 2015a,b; Choi *et al*, 2016; Goto-Ito *et al*, 2018; Karki *et al*, 2018; Lin *et al*, 2018).

Deletion of meA did not affect the total levels of PTPδ or IL1RAPL1 proteins in the mouse brain (Fig 4E and F). In contrast, the pTyr levels of IL1RAPL1 proteins were substantially decreased (Fig 4F), as revealed by immunoprecipitation of IL1RAPL1 from whole-brain lysates followed by immunoblotting for pTyr. This change was associated with the decreases in the levels of IL1RAPL1 in synaptic fractions (SPM and PSD, although not P2) (Fig 4G). In contrast, levels of Slitrk3, IL1RAcP, and NGL-3 in synaptic fractions (P2, SPM, and PSD) were comparable to those in WT mice (Fig EV3H). These biochemical results from *Ptprd-meA*$^{-/-}$ mice recapitulate those observed in *Ptprd*$^{-/-}$ mice (Fig 4A and B). Notably, PTPδ levels were also decreased in the PSD (but not P2 and SPM) fraction of *Ptprd-meA*$^{-/-}$ mice (Fig 4G), likely reflecting the loss of PTPδ variants without meA from the presynaptic nerve terminals.

In line with these biochemical results, the SLM layer in the *Ptprd-meA*$^{-/-}$ hippocampus exhibited significantly decreased excitatory synaptic transmission, as shown by a decreased I/O ratio, but

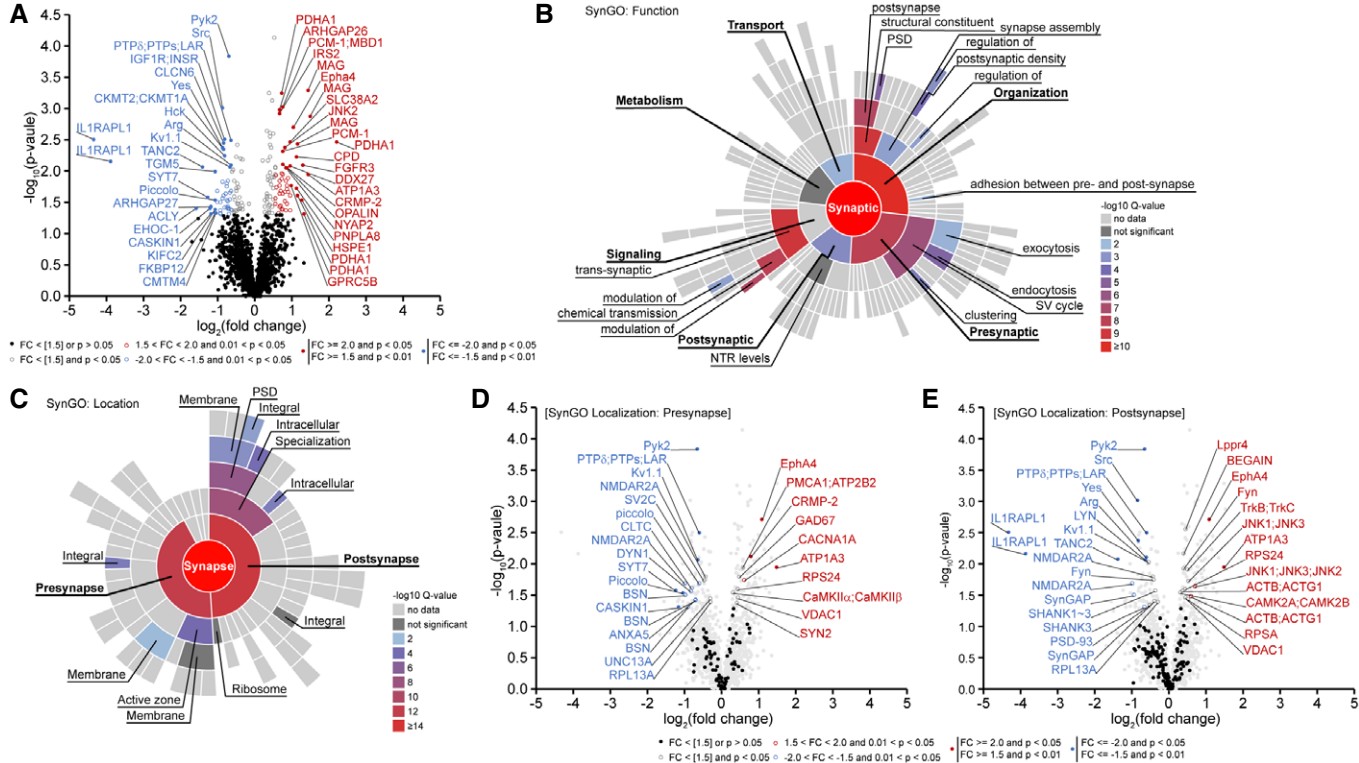

**Figure 3. Altered pTyr levels in multiple pre- and postsynaptic proteins in the *Ptprd*⁻/⁻ brain.**

A   A volcano plot of proteins that showed significant up- or downregulation of pTyr levels (absolute fold change > 1.5, *P* < 0.05) in the whole brain of *Ptprd*⁻/⁻ mice (P21). Note that IL1RAPL1 is the protein with the strongest decrease in pTyr levels (*n* = 3 for WT and 3 for KO). See Dataset EV1, "PhosphoScan Results" tab for a full list of significantly changed proteins.

B   SynGO analysis (https://www.syngoportal.org/) of the 167 proteins that show significant (*P* < 0.05; fold changes were not considered for maximal inclusion of the proteins) up- or downregulation of pTyr levels in *Ptprd*⁻/⁻ mice (P21) for their synaptic functions. See Dataset EV1, "SynGO Ontologies" and "SynGO Annotation" tabs for a full list of SynGO annotations and genes.

C   SynGO analysis of the proteins with significant (*P* < 0.05) up- or downregulation of pTyr levels at their pre- and postsynaptic localizations.

D   A volcano plot of presynaptic proteins included in the SynGO list to show that they are significantly changed in pTyr levels (*P* < 0.05).

E   A volcano plot of postsynaptic proteins included in the SynGO list to show that they are significantly changed in pTyr levels (*P* < 0.05).

Source data are available online for this figure.

---

normal paired-pulse ratio (Fig 4H and I), again recapitulating the results from *Ptprd*⁻/⁻ mice (Fig 2I and J).

These shared phenotypes of *Ptprd*⁻/⁻ and *Ptprd-meA*⁻/⁻ mice collectively suggest that alternative splicing of PTPδ regulates the trans-synaptic interaction of PTPδ with IL1RAPL1, the synaptic localization of IL1RAPL1, and excitatory synapse development and strength. In addition, the global decrease in synaptic levels of IL1RAPL1 protein in the whole brain (including the hippocampus) of both *Ptprd*⁻/⁻ and *Ptprd-meA*⁻/⁻ mice suggests widespread impacts of the loss of splicing-dependent PTPδ–IL1RAPL1 interactions.

**Shared hyperactivity and sleep disturbance phenotypes in *Ptprd*⁻/⁻, *Emx1-Cre;Ptprd*fl/fl, and *Ptprd-meA*⁻/⁻ mice**

PTPδ has been linked to diverse brain disorders, including ADHD, restless leg syndrome, bipolar disorder, obsessive–compulsive disorder, and intellectual disability (Uhl & Martinez, 2019). Because *Ptprd*⁻/⁻ and *Ptprd-meA*⁻/⁻ mice show shared biochemical and synaptic phenotypes, including decreased synaptic localization of

IL1RAPL1 and excitatory synaptic strength, we tested whether *Ptprd* deletion in mice leads to behavioral abnormalities and, if so, whether *Ptprd*⁻/⁻ and *Ptprd-meA*⁻/⁻ mice display shared phenotypes.

We first subjected mice to the Laboras test, where mouse behaviors are continuously monitored for several days in Laboras cages, representing a familiar environment. A quantitative analysis of distance moved and immobility duration during light-off (active) and light-on (inactive) periods (Fig 5A) showed that *Ptprd*⁻/⁻ mice exhibited strong hyperactivity and concomitantly decreased immobility during light-off periods as well as during the first 3–6 h of light-on periods (Fig 5B and C; Appendix Fig S1A and B), suggestive of sleep dysfunction.

These changes were strongly recapitulated by conditional KO (cKO) of *Ptprd* restricted to glutamate neurons (*Emx1-Cre;Ptprd*fl/fl; embryonic) (Gorski *et al*, 2002), but much less so by *Ptprd* cKO in GABAergic neurons (*Viaat-Cre;Ptprd*fl/fl; embryonic) (Chao *et al*, 2010), and not at all by *Ptprd* cKO in postnatal forebrain glutamate neurons (*Camk2a-Cre;Ptprd*fl/fl) (Tsien *et al*, 1996) (Fig 5B and C; Appendix Fig S1C–H). These results suggest that PTPδ expression in

glutamatergic neurons is more important for locomotor and sleep behaviors.

*Ptprd*$^{-/-}$ mice displayed modest hyperactivity in the open-field apparatus, representing a novel environment, a behavior mimicked by *Emx1-Cre;Ptprd*$^{fl/fl}$ mice, but not by *Viaat-Cre;Ptprd*$^{fl/fl}$ or *Camk2a-Cre;Ptprd*$^{fl/fl}$ mice (Fig EV4A and B). *Ptprd*$^{-/-}$ mice displayed mixed anxiety-like behaviors, showing normal behavior in the open-field center region, but moderately anxiolytic behavior in the elevated plus-maze and anxiety-like behavior in the light–dark apparatus; all of these behaviors were more closely mimicked by *Emx1-Cre;Ptprd*$^{fl/fl}$ mice compared with *Viaat-Cre;Ptprd*$^{fl/fl}$ and *Camk2a-Cre;Ptprd*$^{fl/fl}$ mice (Fig EV4C–G).

Importantly, *Ptprd-meA*$^{-/-}$ mice in Laboras cages displayed strong hyperactivity during light-off periods as well as during the first 6 h of light-on periods (Fig 5D and E; Appendix Fig S1I and J), similar to the results from *Ptprd*$^{-/-}$ mice. *Ptprd-meA*$^{-/-}$ mice were modestly hyperactive in the open-field test (Fig EV4A and B), which, together with the Laboras results, largely mimic the behaviors of *Ptprd*$^{-/-}$ mice. In addition, *Ptprd-meA*$^{-/-}$ mice showed normal anxiety-like behavior in the open-field test and anxiolytic-like behavior in the elevated plus-maze test, similar to *Ptprd*$^{-/-}$ mice, but normal anxiety-like behavior in the light–dark test, although there was a tendency for an increase (Fig EV4C–G), similar to *Ptprd*$^{-/-}$ mice.

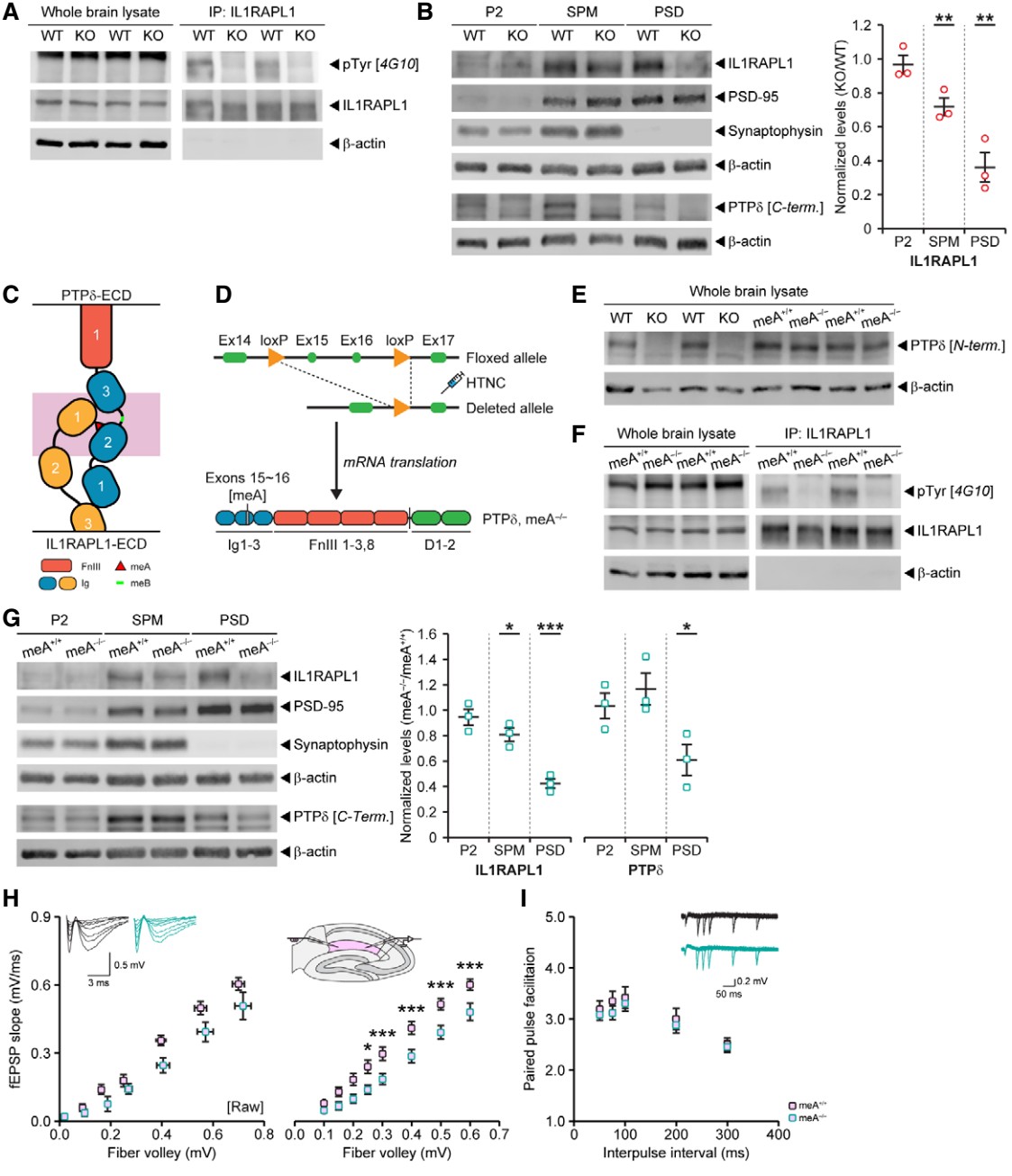

**Figure 4.**

**Figure 4.** PTPδ trans-synaptically regulates tyrosine phosphorylation and synaptic localization of IL1RAPL1 and excitatory synaptic strength.

A Decreased pTyr levels, but normal total levels of IL1RAPL1 protein, in the $Ptprd^{-/-}$ brain (P21–27), as revealed by immunoprecipitation of IL1RAPL1 from whole-brain lysates followed by immunoblotting for total IL1RAPL1 protein and pTyr levels (4G10 antibody).

B Decreased levels of synaptic IL1RAPL1 protein in the $Ptprd^{-/-}$ brain (P21–27), as revealed by immunoblotting of crude synaptosomal (P2), synaptic plasma membrane (SPM), and PSD (PSD II) fractions. Analysis of immunoblots reveals significant decrease of IL1RAPL1 in the SPM and PSD fractions of $Ptprd^{-/-}$ brain samples ($n$ = 3 mice [WT and KO] for each fraction [P2, SPM, and PSD], mean ± SEM, **$P$ < 0.01, Student's $t$-test).

C Schematic diagram showing that the PTPδ-meA splice insert is important for the interaction between PTPδ and IL1RAPL1. Note that the meA splice insert in the Ig2 domain of PTPδ interacts with the Ig1 domain of IL1RAPL1.

D Schematic diagram showing the strategy for generating mice carrying a deletion of the PTPδ-meA splice insert encoded by exons 15 and 16 ($Ptprd$-$meA^{-/-}$).

E Normal levels of the PTPδ protein in whole-brain lysates of $Ptprd$-$meA^{-/-}$ mice (P21–27), which contrasts with the $Ptprd^{-/-}$ brain, where PTPδ protein is undetectable.

F Strongly decreased levels of tyrosine-phosphorylated PTPδ protein in whole-brain lysates of $Ptprd$-$meA^{-/-}$ mice (P21–27), as shown by the immunoprecipitation of IL1RAPL1 protein from whole-brain lysates followed by immunoblotting for total and tyrosine-phosphorylated PTPδ protein.

G Decreased levels of synaptic IL1RAPL1 protein in the $Ptprd$-$meA^{-/-}$ brain (P21–27), as revealed by immunoblotting P2, SPM, and PSD (PSD II) fractions. Analysis of immunoblots reveals significant decrease of IL1RAPL1 in the SPM and PSD fractions of $Ptprd$-$meA^{-/-}$ brain samples ($n$ = 3 mice [$Ptprd$-$meA^{+/+}$ and $Ptprd$-$meA^{-/-}$] and [P2, SPM, and PSD], mean ± SEM, *$P$ < 0.05, ***$P$ < 0.001, Student's $t$-test). Note that, while unchanged in the P2 and SPM fractions, PTPδ is significantly decreased in the PSD of $Ptprd$-$meA^{-/-}$ samples.

H Decreased basal excitatory synaptic transmission in the hippocampal SLM region of $Ptprd$-$meA^{-/-}$ mice, as shown by the I/O ratio at the TA-SLM pathway (P20–24) ($n$ = 15 slice from 5 mice [$Ptprd$-$meA^{+/+}$], 12, 4 [$Ptprd$-$meA^{-/-}$], mean ± SEM, *$P$ < 0.05; ***$P$ < 0.001, two-way RM ANOVA with Holm-Sidak test).

I Normal presynaptic release in the hippocampal SLM region of $Ptprd$-$meA^{-/-}$ mice, as shown by the PPF ratio at the TA-SLM pathway (P20–24) ($n$ = 15 slice from 5 mice [$Ptprd$-$meA^{+/+}$] and 13, 4 [$Ptprd$-$meA^{-/-}$], mean ± SEM, ns, not significant, two-way RM ANOVA).

Source data are available online for this figure.

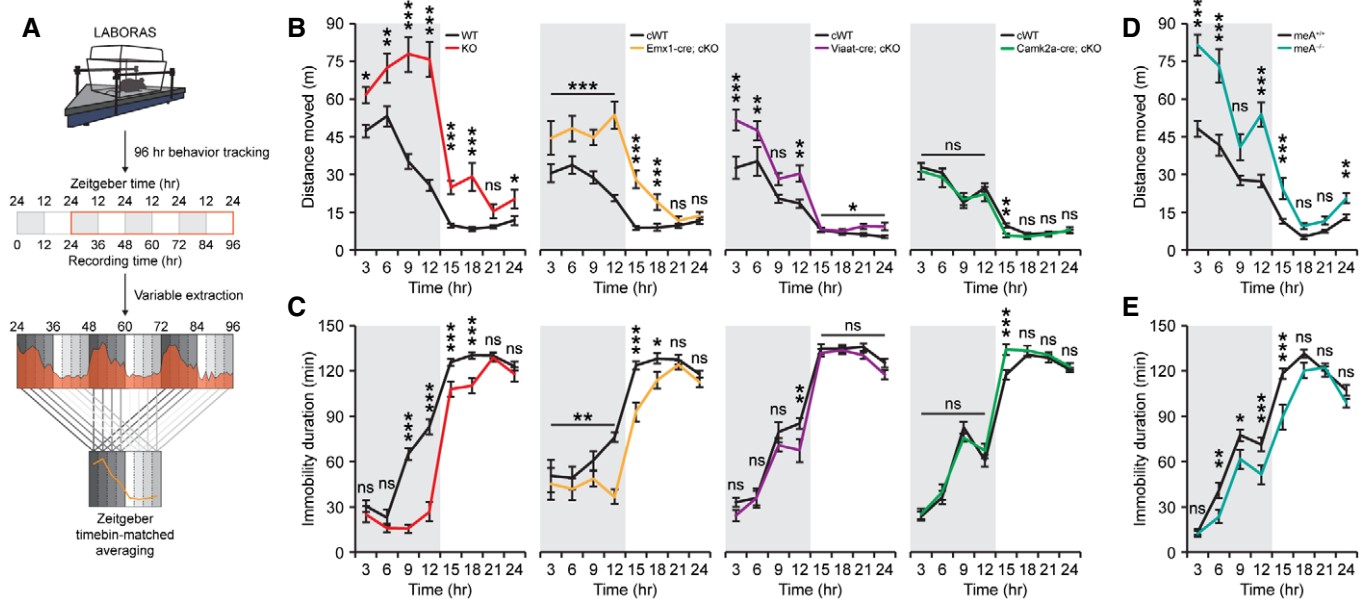

**Figure 5.** Shared hyperactivity and sleep disturbance phenotypes in $Ptprd^{-/-}$, $Emx1$-$Cre;Ptprd^{fl/fl}$ and $Ptprd$-$meA^{-/-}$ mice.

A Schematic diagram showing how mouse movements are measured in Laboras cages for four consecutive days and analyzed. Data from the first 24 h, when the mouse is not fully familiarized with the environment, were not included in the analysis. Each 24 h was sub-divided into 3-h time bins and variables averaged across matching time bins into a representative 24-h period.

B, C Hyperactivity of $Ptprd^{-/-}$ mice (2.5–4 months) in Laboras cages observed during light-off and light-on periods was strongly mimicked by $Emx1$-cKO mice, but more weakly by $Viaat$-cKO mice and not at all by $Camk2a$-cKO mice. Note also that immobility, a measure that better reflects sleep behavior, showed similar changes among different mouse lines ($n$ = 14 mice [WT], 14 [global KO], 16 [$Emx1$-cWT], 16 [$Emx1$-cKO], 11 [$Viaat$-cWT], 13 [$Viaat$-cKO], 18 [$Camk2a$-cWT] and 13 [$Camk2a$-cKO], mean ± SEM, *$P$ < 0.05, **$P$ < 0.01, ***$P$ < 0.001; ns, not significant, two-way RM ANOVA with Holm-Sidak test).

D, E Hyperactive behavior and decreased immobility of $Ptprd$-$meA^{-/-}$ mice (2.5–4 months) in Laboras cages observed during light-off periods and light-on periods, similar to $Ptprd^{-/-}$ mice ($n$ = 7 mice for $Ptprd$-$meA^{+/+}$ and 7 for $Ptprd$-$meA^{-/-}$, mean ± SEM, *$P$ < 0.05, **$P$ < 0.01, ***$P$ < 0.001, ns, not significant, two-way RM ANOVA with Holm-Sidak test).

These results indicate that hyperactive behavior during light-off and light-on periods in Laboras cages is highly similar among $Ptprd^{-/-}$, $Emx1$-$Cre;Ptprd^{fl/fl}$, and $Ptprd$-$meA^{-/-}$ mice, and that

anxiety-like behaviors are partly similar among $Ptprd^{-/-}$, $Emx1$-$Cre$; $Ptprd^{fl/fl}$, and $Ptprd$-$meA^{-/-}$ mice. In addition, the shared hyperactivity and sleep disturbances in the three mouse lines strongly

suggest that the PTPδ–IL1RAPL1 interaction is important for the neural circuits that control locomotor and sleep behaviors.

### Shared decrease in non-REM (rapid eye movement) sleep duration and delta power in *Emx1-Cre;Ptprd^{fl/fl}* and *Ptprd-meA^{−/−}* mice

One intriguing behavioral phenotype shared by *Ptprd^{−/−}*, *Emx1-Cre;Ptprd^{fl/fl}*, and *Ptprd-meA^{−/−}* mice was the increased locomotor activity and decreased immobility observed during the first 3–6 h of light-on periods (Fig 5). Does this imply disrupted sleep behavior? Because sleep is important for normal brain function and is disrupted in various neuropsychiatric disorders (Hobson & Pace-Schott, 2002), and *PTPRD* has been implicated in restless leg syndrome (Schormair *et al*, 2008), we further explored this phenotype by performing 24-h EEG (electroencephalogram) and EMG (electromyogram) recordings to determine whether mice are in WAKE, NREM (non-rapid eye movement sleep), or REM (rapid eye movement sleep) states (Fig 6A). We first used *Emx1-Cre;Ptprd^{fl/fl}* mice for these analyses because they carry a *Ptprd* deletion in cortical glutamatergic neurons, known to be critical for cortical EEG rhythms (Hobson & Pace-Schott, 2002).

During the first 3 h of the light-on (inactive) period, control mice showed a normal decrease in the duration of WAKE states and a normal increase in the duration of NREM sleep states, as determined by EEG and EMG patterns (Figs 6B and EV5). In contrast, *Emx1-Cre;Ptprd^{fl/fl}* mice showed an abnormally increased WAKE duration and decreased NREM duration, compared with control mice. Furthermore, these changes were associated with decreased delta power (representing the strength of deep sleep oscillations) in NREM oscillations but not in WAKE oscillations, as shown by normalized (to total power) data (Fig 6C). REM sleep was unaffected, and correspondingly, the *Emx1-Cre;Ptprd^{fl/fl}* mice displayed normal theta power during REM sleep.

Importantly, *Ptprd-meA^{−/−}* mice also displayed changes in NREM sleep duration and NREM delta power that are similar to those observed in *Ptprd^{−/−}* mice. Specifically, *Ptprd-meA^{−/−}* mice displayed increased WAKE duration and decreased NREM duration in the first 3 h of the light-on phase (Fig 6D) and decreased delta power during NREM sleep (Fig 6E). Certain changes were unique to *Ptprd-meA^{−/−}* mice, which included changed WAKE and NREM durations during light-off periods (in addition to light-on periods) and decreased delta power in WAKE and NREM oscillations as well as decreased theta power in REM oscillations.

These results collectively suggest that *Emx1-Cre;Ptprd^{fl/fl}* and *Ptprd-meA^{−/−}* mice show shared abnormalities in sleep rhythms, indicated by decreases in NREM sleep duration and NREM delta power during the first few hours of the light-on period. Again, these shared abnormalities between *Emx1-Cre;Ptprd^{fl/fl}* and *Ptprd-meA^{−/−}* mice suggest that the PTPδ–IL1RAPL1 interaction is important for normal sleep rhythms.

## Discussion

Our data indicate that endogenous PTPδ, labeled by *in vivo* tagging, is expressed in various brain regions and is present at excitatory presynaptic sites in the hippocampus. Functionally, PTPδ is required for normal excitatory synapse density and strength in the hippocampal SLM layer. In addition, PTPδ deletion induces changes in pTyr levels in synaptic proteins, including a strong decrease in IL1RAPL1. PTPδ deletion also leads to abnormal sleep behavior and rhythms. Importantly, selective deletion of the PTPδ-meA insert, critical for the PTPδ–IL1RAPL1 interaction, recapitulates the biochemical, synaptic, and sleep phenotypes caused by global or conditional PTPδ deletion.

Perhaps, the most important conclusion of our study is that presynaptic PTPδ trans-synaptically interacts *in vivo* with postsynaptic IL1RAPL1 through the meA splice insert to regulate excitatory synapse density and sleep behavior and rhythms. This conclusion is supported by multiple lines of *in vivo* evidence, including (i) decreased excitatory synaptic strength in the hippocampal SLM layer shared by *Ptprd^{−/−}* and *Ptprd-meA^{−/−}* mice, (ii) decreased pTyr levels and synaptic localization of IL1RAPL1 shared by *Ptprd^{−/−}* and *Ptprd-meA^{−/−}* mice, and (iii) abnormal sleep behavior and rhythms shared by *Emx1-Cre;Ptprd^{fl/fl}* and *Ptprd-meA^{−/−}* mice.

Previous studies on *in vivo* functions of splice inserts in synaptic adhesion molecules have largely focused on complexes of neurexins and neuroligins. These studies have revealed a role for splice site inserts in determining the specificity and affinity of binding partners, localization to excitatory versus inhibitory synapses, and the magnitude of postsynaptic responses mediated AMPARs and NMDARs (Boucard *et al*, 2005; Chih *et al*, 2006; Graf *et al*, 2006; Aoto *et al*, 2013; Dai *et al*, 2019). Our study extends these findings by demonstrating that alternative splicing regulates the "density" of excitatory synapses by promoting a particular trans-synaptic interaction, in this case, the interaction of PTPδ with IL1RAPL1, but not with other postsynaptic partners of PTPδ (NGL-3, Slitrks, IL-1RAcP, and SALM3/5).

Our EM results from PTPδ-tdTomato mice suggest that PTPδ is mainly present at presynaptic sites in the adult hippocampus, while it is localized both pre- and postsynaptically in immature cultured neurons (Fig 1D and F). These results directly support the presynaptic localization of PTPδ as well as other LAR-RPTPs (LAR and PTPσ) predicted to be presynaptic based on cell biological results, for example, mixed-culture synapse formation assays (Woo *et al*, 2009; Kwon *et al*, 2010; Takahashi *et al*, 2011, 2012; Valnegri *et al*, 2011; Yoshida *et al*, 2011, 2012; Yim *et al*, 2013; Li *et al*, 2015; Choi *et al*, 2016). However, other studies have suggested postsynaptic localization and functions of PTPδ, including in the regulation of dendritic arborization (Nakamura *et al*, 2017) and surface expression of AMPARs (Dunah *et al*, 2005).

Data from PTPδ-tdTomato mice also suggest that PTPδ is present at excitatory (but not inhibitory) presynaptic sites, at least in the hippocampus, a conclusion supported by the decreased excitatory, but not inhibitory, synapse density in *Ptprd^{−/−}* mice, and by the shared decreases in excitatory synaptic strength in the SLM layer in *Ptprd^{−/−}* and *Ptprd-meA^{−/−}* mice (Fig 2). Notably, previous *in vitro* studies have suggested PTPδ functions at both excitatory and inhibitory synaptic sites (Takahashi & Craig, 2013; Um & Ko, 2013), as reflected in the fact that PTPδ, but not other LAR-RPTPs, binds to the inhibitory synaptic organizer, Slitrk3 (Takahashi *et al*, 2012; Yim *et al*, 2013). However, other postsynaptic partners of PTPδ have been shown to promote either excitatory or both excitatory and inhibitory synapse development, as follows: Slitrk2, excitatory and inhibitory; Slitrk1/4/5, excitatory (Takahashi *et al*, 2012); SALM3/5, excitatory and inhibitory; IL1RAPL1, excitatory; IL1RAcP, excitatory; and NGL-3, excitatory (Woo *et al*, 2009; Kwon *et al*, 2010; Yoshida *et al*, 2011,

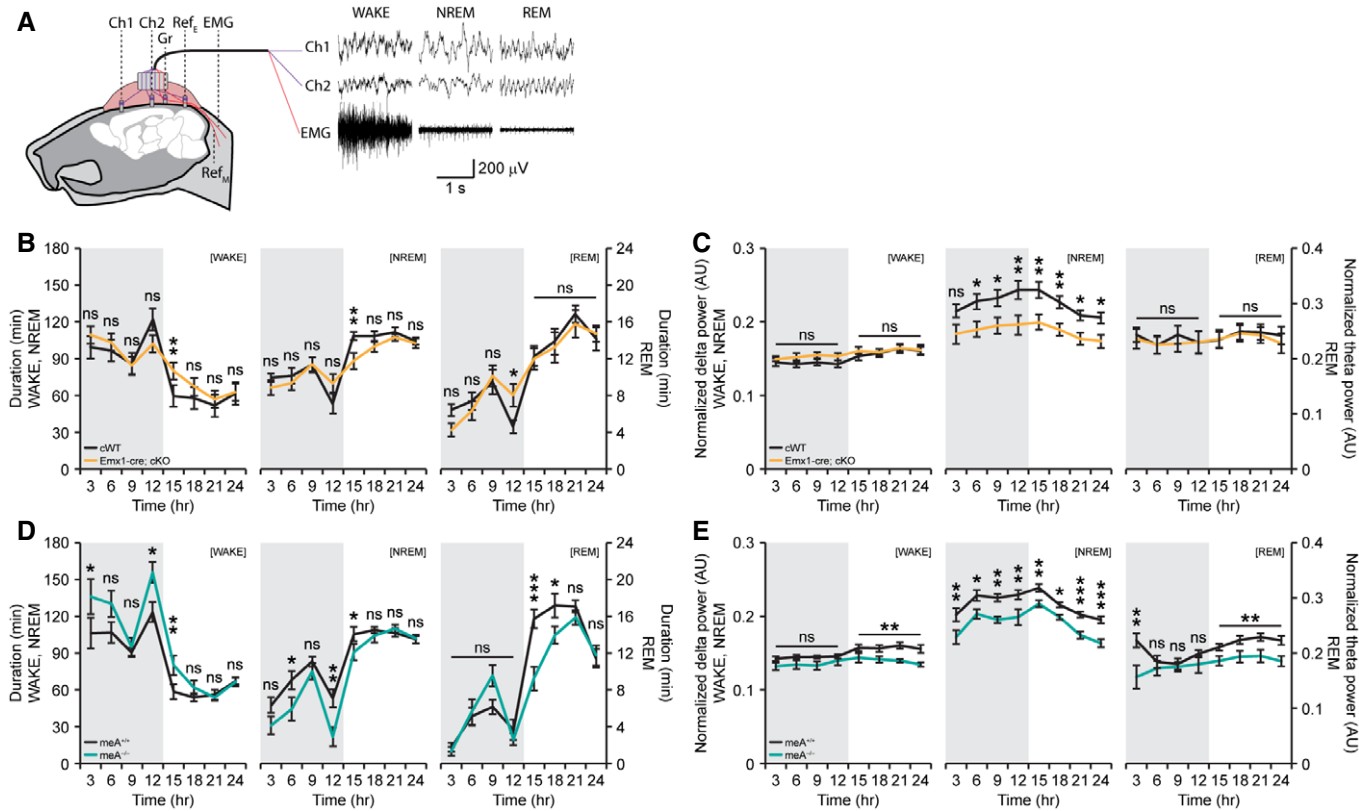

**Figure 6. Shared decreased NREM sleep and delta power phenotypes in *Emx1-Cre;Ptprd*$^{fl/fl}$ and *Ptprd-meA*$^{-/-}$ mice.**

A   Schematic diagram showing the sites of electroencephalography (EEG) and electromyography (EMG) surgery (frontal [Ch1], parietal [Ch2], EEG reference [Ref$_E$], trapezius muscle [EMG], and EMG reference [Ref$_M$] probes), and examples of EEG and EMG traces during WAKE, NREM, and REM states. Gr, animal ground.

B   Increased WAKE duration and decreased NREM duration during the first 3 h of the light-on period in *Emx1-Cre;Ptprd*$^{fl/fl}$ mice (2.5–4 months) (*n* = 9 mice [*Emx1*-cWT] and 9 mice [*Emx1*-cKO], mean ± SEM, *$P$ < 0.05, **$P$ < 0.01, ns, not significant, two-way RM ANOVA with Holm-Sidak test).

C   Decreased delta power (normalized to total power) in NREM, but not WAKE, oscillations in *Emx1-Cre;Ptprd*$^{fl/fl}$ mice (2.5–4 months). Note also the normal theta power (normalized to total power) in REM oscillations in *Emx1-Cre;Ptprd*$^{fl/fl}$ mice (*n* = 9 [*Emx1*-cWT] and 9 [*Emx1*-cKO], mean ± SEM, *$P$ < 0.05, **$P$ < 0.01, ns, not significant, two-way RM ANOVA with Holm-Sidak test).

D   Increased WAKE duration and decreased NREM duration during the first 3 h of the light-on period and decreased REM duration in the first half of the light-on period in *Ptprd-meA*$^{-/-}$ mice (2.5–4 months). Note also that *Ptprd-meA*$^{-/-}$ mice also display increased WAKE duration and decreased NREM duration during the light-off period (*n* = 8 [*Ptprd-meA*$^{+/+}$] and 8 [*Ptprd-meA*$^{-/-}$], mean ± SEM, *$P$ < 0.05, **$P$ < 0.01, ***$P$ < 0.001, ns, not significant, two-way RM ANOVA with Holm-Sidak test).

E   Decreased delta power (normalized to total power) in NREM (both light-off and light-on periods) and WAKE (light-on period) oscillations, and decreased theta power in REM (mainly light-on period) oscillations in *Ptprd-meA*$^{-/-}$ mice (2.5–4 months) (*n* = 8 [*Ptprd-meA*$^{+/+}$] and 8 [*Ptprd-meA*$^{-/-}$], mean ± SEM, *$P$ < 0.05, **$P$ < 0.01, ***$P$ < 0.001, ns, not significant, two-way RM ANOVA with Holm-Sidak test).

2012; Li *et al*, 2015; Choi *et al*, 2016). Our PTPδ-tdTomato reporter mice may serve as a useful tool for determining whether PTPδ is present and functions at excitatory or inhibitory synapses, or at pre- or postsynaptic sites, in other brain areas.

Our study provides a list of synaptic and non-synaptic proteins with altered pTyr levels in the whole brain of *Ptprd*$^{-/-}$ mice (Fig 3). While these pre- and postsynaptic molecules with altered pTyr levels might contribute to PTPδ-dependent regulation of excitatory synapse development, our data suggest a strong contribution by IL1RAPL1, a postsynaptic adhesion molecule reported to be important for excitatory synapse development and maintenance both *in vitro* and *in vivo* (Pavlowsky *et al*, 2010a,b, Valnegri *et al*, 2011; Yasumura *et al*, 2014; Montani *et al*, 2017). The interaction of PTPδ with IL1RAPL1 critically requires the meA miniexon (Yoshida *et al*, 2011; Yamagata *et al*, 2015b), unlike the interactions with other postsynaptic partners (NGL-3, SALM3/5, TrkC, Slitrks, and

IL1RAcP), allowing us to surgically disrupt this trans-synaptic inter-action via meA deletion without changing total PTPδ levels. Indeed, both *Ptprd*$^{-/-}$ and *Ptprd-meA*$^{-/-}$ mice displayed decreased excita-tory synaptic strength, decreased IL1RAPL1-pTyr levels, and abnor-mal sleep behavior and rhythms. Because IL1RAPL1 is the only known postsynaptic binding partner of PTPδ that requires meA for PTPδ binding, we hypothesize that PTPδ-dependent regulation of excitatory synapse development and strength is strongly IL1RAPL1-dependent.

Strong additional support for this inference is provided by the observation that synaptic levels of IL1RAPL1 protein are robustly decreased in both *Ptprd*$^{-/-}$ and *Ptprd-meA*$^{-/-}$ mice (Fig 4A–G). How might this change lead to decreased excitatory synapse density? The trans-synaptic PTPδ–IL1RAPL1 interaction is known to promote lateral interactions between neighboring PTPδ–IL1RAPL1 complexes, leading to higher-order trans-synaptic assembly and

strong synaptogenesis (Won *et al*, 2017). These mechanisms could be suppressed in our PTPδ-mutant mice, which seems to be further supported by the decreased synaptic levels of PTPδ in *Ptprd-meA*$^{-/-}$ mice (Fig 4G). IL1RAPL1 also interacts with PSD-95 through its cytoplasmic tail (Pavlowsky *et al*, 2010b), although synaptic levels of PSD-95 were not decreased in our PTPδ-mutant mice. The PTPδ–IL1RAPL1 interaction also promotes IL1RAPL1 binding to RhoGAP2 (Rho GTPase-activating protein) and Mcf2 l (Rho guanine nucleotide exchange factor), which are known to regulate excitatory synapse density and AMPAR surface trafficking (Valnegri *et al*, 2011; Hayashi *et al*, 2013), suggesting the possibility of altered Rho signaling in PTPδ-mutant mice.

PTPδ has been implicated in various brain disorders, including ADHD, restless leg syndrome, addiction, bipolar disorder, obsessive–compulsive disorder, and intellectual disability (Anney *et al*, 2008; Schormair *et al*, 2008; Elia *et al*, 2010; Distel *et al*, 2011; Malhotra *et al*, 2011; Yang *et al*, 2011; Kim *et al*, 2013; Jarick *et al*, 2014; Moore *et al*, 2014; Choucair *et al*, 2015; Drgonova *et al*, 2015; Mattheisen *et al*, 2015; Uhl *et al*, 2018; Uhl & Martinez, 2019). Our results indicate that hyperactivity and sleep disturbances are shared by *Ptprd*$^{-/-}$, *Emx1-Cre;Ptprd*$^{fl/fl}$, and *Ptprd-meA*$^{-/-}$ mice (Fig 5). In addition, NREM delta powers are strongly decreased in *Emx1-Cre; Ptprd*$^{fl/fl}$ and *Ptprd-meA*$^{-/-}$ mice (Fig 6). NREM delta power is known to indicate sleep pressure (Dijk *et al*, 1990; Feinberg *et al*, 2006; Tononi & Cirelli, 2006), which requires normal cortico-cortical connections (Hobson & Pace-Schott, 2002). Therefore, PTPδ deletion in cortical excitatory neurons may contribute to the abnormal sleep behavior and rhythms in our PTPδ-mutant mice by altering cortico-cortical synapses through the loss of IL1RAPL1. In line with this possibility, synaptic levels of IL1RAPL1 protein are strongly decreased in the whole brain of *Ptprd*$^{-/-}$ and *Ptprd-meA*$^{-/-}$ mice (Fig 4).

In conclusion, our data provide *in vivo* evidence suggesting that alternative splicing in presynaptic PTPδ selectively regulates the trans-synaptic interaction of PTPδ with postsynaptic IL1RAPL1 and excitatory synapse development as well as sleep behavior and rhythms in mice. More generally, our study demonstrates that alternative splicing can regulate synapse development and sleep *in vivo*.

# Materials and Methods

### Animals

Generation of the LoxP-flanked *Ptprd* mutant mice, serviced by Biocytogen, was achieved by the introduction of a targeting vector containing exon 13 (NCBI reference sequence NM_011211.4) of the *Ptprd* gene (ENSMUSE00000466242) flanked by loxP sites and an En2-SA-IRES-EGFP-PA-loxP-neomycin (henceforth "neomycin") cassette flanked by Frt sites (KO-first, conditional-ready strategy) was introduced to C57BL/6J ES cells by homologous recombination. Targeted ES cells were injected into BALB/c blastocysts to generate chimeric mice, which were crossed with C57BL/6J mice to generate F1 progeny. F1 mice were crossed with C57BL/6J mice for three generations to ensure background purity. The EGFP in the neomycin failed to express due to weak promoter activity. F4 heterozygous mice were bred with Protamine-Flp mice (C57BL/6J) to remove the neomycin cassette, with the resulting "floxed" mice bred and maintained in a homozygote state. Whole-body knockout (*Ptprd*$^{-/-}$) mice were obtained by first injection of purified hexa-histidine-TAT-NLS-Cre (HTNC) enzyme into the two-cell heterozygous floxed embryo as a one-step method of acquiring *Ptprd*$^{+/-}$ (HT) mice, removing the need to outbreed genetically coded *Cre*, as reported recently (Shin *et al*, 2019). The resulting HT mice were mated in HT × HT fashion to obtain *Ptprd*$^{+/+}$ (WT) and *Ptprd*$^{-/-}$ mice. To obtain conditional deletion mice, homozygous floxed mice were mated with mice containing one allele of the floxed *Ptprd* mutant and one allele of the desired *Cre* gene. Of the resulting pups, homozygous floxed mice with the *Cre* transgene were used as conditional KOs (cKOs), while those without the *Cre* transgene were used as controls. For genotyping, following primers for the neomycin cassette were used: Flp_Forward [GTGGATCGATCCTACCCCT TGCG], Flp_Reverse [GGTCCAACTGCAGCCCAAGCTTCC]. Primers for the loxP sites were as follows: LoxP_Forward [GGACCTTGAC-CAAAACAACCC], LoxP_Reverse [GAGGGAGTCTATCTCATAAAA GC]. Deletion of exon 3 by cre recombination was confirmed by the addition of Del_Forward [GACTGTGCTCCACAACTCTG] to the LoxP PCR, which results in a ~180-bp band only present with the deletion of exon 3. Primers for the cre recombinase were as follows: Cre_Forward [GTGTTGCCGCATCTGC], Cre_Reverse [CACCATTGCCCCT GTTTCACTATC].

To generate the PTPδ-linker-tdTomato fusion protein in mice, serviced by Biocytogen, sgRNAs were designed to target the region between exon31 and 3′UTR. The linker (GSAGSAAGSGEF) is based on published literature (Waldo *et al*, 1999). For the targeting site, candidate guide RNAs were designed by the CRISPR design tool (http://crispr.mit.edu). The gene-targeting vector containing linker-tdTomato and 2 homology arms of left (1,435 bp) and right (1,289 bp) each was used as a template to repair the DSBs generated by Cas9/sgRNA. The linker-tdTomato reporter was precisely inserted before stop codon of the *PTPRD* gene. C57BL/6N female mice and KM mouse strains were used as embryo donors and pseudopregnant foster mothers, respectively. Different concentrations of Cas9 mRNA and sgRNA were mixed and co-injected into the cytoplasm of one-cell stage fertilized eggs. After injection, surviving zygotes were transferred into oviducts of KM albino pseudopregnant females. Founder pups were positively confirmed by PCR product screening. F1 heterozygous mice were obtained, after which subsequent generations were bred with C57BL/6J mice for at least 4 generations to ensure background purity. Starting with F5, a subset of mutants heterozygous × heterozygous mating was used to obtain homozygous mice, and subsequent generations were thereafter maintained in the homozygous state. For genotyping the PTPδ:tdTomato fusion reporter mice, following primers were used: Tomato_Control_Forward [CCGAAAATCTGTGGGAAGTC], Tomato_Control_Reverse [AAGGGAGCTGCAGTGGAGTA], Tomato_Mutant_Forward [GGCATTAAAGCAGCGTATCC], Tomato_Mutant_Reverse [CTGTTCCTGTACGGCATGG].

To generate *Ptprd-meA*$^{-/-}$ mice, serviced by Biocytogen, sgRNAs were designed to targets the regions flanking the region containing exons 15 and 16 (henceforth "target"). Candidate guide RNAs targeting the 5′ end [GGGTCAAAGCCCTCGAGCATGGG] and 3′ end [GAACCTGTTTCCACGGCTGAAGG] of the flanking region were designed using the CRISPR design tool (http://crispr.mit.edu). 5′ sgRNA 1 [TAGGGTCAAAGCCCTCGAGCAT], 5′ sgRNA 2 [AAACA TGCTCGAGGGCTTTGAC], 3′ sgRNA 1 [TAGGAACCTGTTTCCACG GCTGA], 3′ sgRNA 2 [AAACTCAGCCGTGGAAACAGGTT]. The

gene-targeting vector containing the target region flanked by LoxP sequences and 2 homology arms of the 5′ (1250 bp) and 3′ (1330 bp) each was used as a template to repair the two DSBs generated by Cas9/sgRNA. C57BL/6N female mice and KM mouse strains were used as embryo donors and pseudopregnant foster mothers, respectively. Different concentrations of Cas9 mRNA and sgRNA were mixed and co-injected into the cytoplasm of one-cell stage fertilized eggs. After injection, surviving zygotes were transferred into oviducts of KM albino pseudopregnant females. Founder pups were positively confirmed by PCR product screening. F1 heterozygous mice were obtained, after which subsequent generations were bred with C57BL/6J mice for at least 4 generations to ensure background purity. Whole-body meA deletion ($Ptprd$-$meA^{-/-}$) mice were obtained by first injection of purified hexa-histidine-TAT-NLS-Cre (HTNC) enzyme into the two-cell LoxP-target-LoxP (floxed) allele heterozygous embryo as a one-step method of acquiring $Ptprd$-$meA^{+/-}$ mice, removing the need to outbreed genetically coded $Cre$ (Shin $et\ al$, 2019).The resulting $Ptprd$-$meA^{+/-}$ mice were mated in $Ptprd$-$meA^{+/-}$ × $Ptprd$-$meA^{+/-}$ fashion to obtain $Ptprd$-$meA^{+/+}$ and $Ptprd$-$meA^{-/-}$ mice. For genotype the floxed and $Ptprd$-$meA^{-/-}$ mice, following primers were used: meA_Forward [TGTCTTAAAAGTCAAAGAATGACTCCCC], meA_Reverse [ATCACTGCTCGAGGACCTCTGGATA], and meA_Mut [GGCCCACACAGTAGCTGTGGCAATA]. All three primers were used in one reaction, which resulted in the following bands: floxed, 396 bp; wild type, 323 bp; mutant, 284 bp.

All animals used in this study were maintained, and procedures performed in accord with the Requirements of Animal Research at KAIST. Experimental procedures were approved by the Committee on Animal Research at KAIST (KA2012-19). Animals were housed in a 13:00 to 01:00 dark/light cycle environment and fed/watered $ad\ libitum$.

### tdTomato imaging

PTPδ-tdTomato mice were intracardially perfused with 4% paraformaldehyde, and each mouse brain was either coronally, sagittally, or horizontally sectioned at 100 μm. Without any staining, it was directly mounted on the slide glass. 10× images were obtained using a slide scanner (Axio Scan.Z1), and 63× magnified images were acquired with a confocal microscope (LSM 780, Zeiss).

### Electron microscopy

$Wild$-$type\ and\ Ptprd$-$knockout$ mice were deeply anesthetized with sodium pentobarbital (80 mg/kg, i.p.) and were intracardially perfused with 10 ml of heparinized normal saline, followed by 50 ml of a freshly prepared fixative of 2.5% glutaraldehyde and 1% paraformaldehyde in 0.1 M phosphate buffer (PB, pH 7.4). Hippocampus was removed from the whole brain, postfixed in the same fixative for 2 h, and stored in PB (0.1 M, pH 7.4) overnight at 4°C. Sections were cut coronally or horizontally on a Vibratome at 50 μm. The sections were osmicated with 1% osmium tetroxide (in 0.1 MPB) for 1 h, dehydrated in graded alcohols, flat-embedded in Durcupan ACM (Fluka), and cured for 48 h at 60°C. Small pieces containing stratum lacunosum moleculare (SLM) of dorsal hippocampal CA1 regions were cut out of the wafers and glued onto the plastic block by cyanoacrylate. Ultrathin sections were cut and mounted on Formvar-coated single-slot grids. For quantification of excitatory synapses, sections were stained with uranyl acetate and lead citrate, and examined with an electron microscope (Hitachi H-7500; Hitachi) at 80 kV accelerating voltage. Twenty-four micrographs representing 368.9 μm² neuropil regions in each mouse were photomicrographed at a 40,000× and used for quantification. The number of spines (PSD density), the proportion of perforated spines, PSD length, and PSD thickness from three WT and $Ptprd$-$knockout$ mice were quantified. The measurements were all performed by an experimenter blind to the genotype. Digital images were captured with GATAN DigitalMicrograph software driving a CCD camera (SC1000 Orius; Gatan) and saved as TIFF files. The brightness and contrast of the images were adjusted in Adobe Photoshop 7.0 (Adobe Systems).

### Postembedding immunogold staining for GABA

Sections were immunostained for GABA by the postembedding immunogold method, as previously described (Paik $et\ al$, 2007) with some modifications. In brief, the grids were treated for 5 min in 1% periodic acid, to etch the resin, and for 8 min in 9% sodium periodate, to remove the osmium tetroxide, then washed in distilled water, transferred to Tris-buffered saline containing 0.1% Triton X-100 (TBST; pH 7.4) for 10 min, and incubated in 2% human serum albumin (HSA) in TBST for 10 min. The grids were then incubated with rabbit antiserum against GABA (GABA 990, 1:10,000) in TBST containing 2% HSA for 2 h at room temperature. The antiserum (a kind gift from professor O. P. Ottersen at the Center for Molecular Biology and Neuroscience, University of Oslo) was raised against GABA conjugated to bovine serum albumin with glutaraldehyde and formaldehyde (Kolston $et\ al$, 1992) and characterized by spot testing (Ottersen & Storm-Mathisen, 1984). To eliminate cross-reactivity, the diluted antiserum was preadsorbed overnight with glutaraldehyde (G)-conjugated glutamate (500 μM, prepared according to Ottersen and Storm-Mathisen (1986)). After extensive rinsing in TBST, grids were incubated for 3 h in goat anti-rabbit IgG coupled to 15 nm gold particles (1:25 in TBST containing 0.05% polyethylene glycol; BioCell Co., Cardiff, United Kingdom). After a rinse in distilled water, the grids were counterstained with uranyl acetate and lead citrate, and examined with an electron microscope (Hitachi H-7500; Hitachi) at 80 kV accelerating voltage. To assess the immunoreactivity for GABA, gold particle density (number of gold particles per μm2) of each GABA$^+$ terminal was compared with gold particle density of terminals which contain round vesicles and make asymmetric synaptic contact with dendritic spines (background density). Terminals were considered GABA-immunopositive (+) if the gold particle density over the vesicle-containing areas was at least five times higher than background density.

### Quantitative analysis of excitatory and inhibitory synapses

For quantification of the distribution patterns of PTPδ-tdTomato signals in neuropils, eight electron micrographs (at 10,000×) representing 1,968.98 μm² neuropil regions in two PTPδ-tdTomato fusion mice were taken. The number of PTPδ-immunopositive presynaptic terminals which that apposed to and not apposed to PSDs, and the number of PTPδ-immunopositive neuropils outside the regions of presynaptic terminals were quantified by using ImageJ software. For

quantification of PSDs in WT and *Ptprd*$^{-/-}$ mice, 24 electron micrographs representing 368.9 μm$^2$ neuropil regions in each mouse were taken at a 40,000×. The number of spines (PSD density), proportion of perforated spines, PSD length, and PSD thickness from each three WT and *Ptprd*$^{-/-}$ mice were quantified by using ImageJ software. For quantification of inhibitory synapses, 24 electron micrographs representing 655.5 μm2 neuropil regions in each mouse were taken at a 30,000×. The number of GABA$^+$ terminals showing clear PSD (inhibitory synapse density), and length and thickness of PSD contacting GABA$^+$ terminals from each three WT and *Ptprd*-knockout mice were quantified by using ImageJ software. The measurements were all performed by an experimenter blind to the genotype. Digital images were captured with GATAN DigitalMicrograph software driving a CCD camera (SC1000 Orius; Gatan) and saved as TIFF files. The brightness and contrast of the images were adjusted in Adobe Photoshop 7.0 (Adobe Systems). For quantification of morphological properties of presynaptic nerve terminals (nerve terminal density, synaptic vesicles density, and cross-sectional area of presynaptic nerve terminals) in WT and *Ptprd*$^{-/-}$ mice, four electron micrographs (at 40,000x) representing 184.5-μm$^2$ neuropil regions in each mouse were taken. Data of each parameters from three WT and *Ptprd*$^{-/-}$ mice were quantified by using ImageJ software.

**Electron microscopic immunohistochemistry**

For immunostaining for tdTomato, tdTomato + GAD65/67, or tdTomato + vGluT2, two PTPδ-tdTomato fusion mice (20–25 g) were used. Animals were deeply anesthetized with sodium pentobarbital (80 mg/kg, i.p.) and were intracardially perfused with 10 ml of heparinized normal saline, followed by 50 ml of a freshly prepared mixture of 4% paraformaldehyde and 0.01% glutaraldehyde in 0.1 M phosphate buffer. Whole brains were removed and postfixed in the same fixative for 2 h at 4°C. Sections were cut horizontally on a Vibratome at 60 μm and cryoprotected in 30% sucrose in PB overnight at 4°C. The sections were frozen on dry ice for 20 mins and thawed in phosphate-buffered saline (PBS, 0.01 M, pH 7.4) to enhance penetration. They were pretreated with 1% sodium borohydride for 30 min to quench glutaraldehyde and then blocked with 3% H$_2$O$_2$ for 10 min to suppress endogenous peroxidases and with 10% normal donkey serum (NDS, Jackson ImmunoResearch, West Grove, PA, USA) for 30 min to mask secondary antibody binding sites. For single immunostaining for tdTomato, sections of brain were incubated in chicken anti-tdTomato antibody (1:5,000) overnight at 4°C; the next day, they were rinsed in PBS for 15 mins and incubated for 2 h in biotinylated donkey anti-chicken (1:200, Jackson ImmunoResearch, West Grove, PA, USA). After washing, the sections were incubated with ExtrAvidin peroxidase (1:5,000, Sigma, St. Louis, MO, USA) for 1 h, and the immunoperoxidase was visualized by nickel-intensified 3,3′-diaminobenzidine tetrahydrochloride (DAB). For double immunostaining for tdTomato and GAD65/67 or tdTomato and vGluT2, sections of the brain pretreated as above were incubated overnight in a mixture of antibodies. Primary antibodies used are as follows: chicken anti-tdTomato (1:5,000), mouse anti-GAD65/67 (1:1,000, MSA-225, Enzo Life Science, MI, USA), and guinea pig anti-vGluT2 (1:400, Af810, Frontier Institute Co. Ltd., Hokkaido, Japan) antibodies. After rinsing in PBS, sections were incubated with a mixture of biotinylated donkey

anti-chicken (1:200) and 1 nm gold-conjugated donkey anti-guinea pig (1:50, EMS, Hatfield, PA, USA) or a mixture of biotinylated goat anti-chicken (1:200) and 1 nm gold-conjugated donkey anti-mouse (1:50, EMS) antibodies for 2 h. The sections were postfixed with 1% glutaraldehyde in PBS for 10 mins, rinsed in PBS several times, incubated for 4 mins with HQ silver enhancement solution (Nanoprobes, Yaphank, NY, USA), and rinsed in 0.1 M sodium acetate and PBS. The sections were incubated with ExtrAvidin peroxidase (1:5000) and visualized with DAB as above. The sections were further rinsed in PB, osmicated (in 1% osmium tetroxide in PB) for 1 h, dehydrated in graded alcohols, flat-embedded in Durcupan ACM (Fluka, Buchs, Switzerland) between strips of Aclar plastic film (EMS), and cured for 48 h at 60°C. Chips containing prominent staining for tdTomato, GAD65/67, or vGluT2 in the brain were cut out of the wafers and glued onto blank resin blocks with cyanoacrylate. Serially cut thin sections were collected on Formvar-coated single-slot nickel grids and stained with uranyl acetate and lead citrate. Grids were examined on a Hitachi H7500 electron microscope (Hitachi, Tokyo, Japan) at 80 kV accelerating voltage. Images were captured with Digital Micrograph software driving a cooled CCD camera (SC1000; Gatan, Pleasanton, CA, USA) attached to the microscope, and saved as TIFF files.

**Mouse primary neuronal culture**

Mouse cultured neurons were prepared from embryonic day 17 fetal PTPδ-tdTomato mice. Briefly, dissected hippocampus and entorhinal cortex tissues were dissociated with papain and plated on poly-D-lysine-coated 18-mm glass coverslips with plating medium (Neurobasal-A medium supplemented with 2% B-27, 2% FBS, 1% GlutaMax, and 1 mM sodium pyruvate, all from Thermo Fisher Scientific). Four hours after plating, the plating medium was replaced with FBS free culture medium (Neurobasal-A medium supplemented with 2% B-27, 1% GlutaMax, and 1 mM sodium pyruvate) and then 50% replacement every 7 days with fresh FBS free culture medium.

**Immunocytochemistry**

DIV 6 or DIV 24 cultured hippocampal/cortical neurons were fixed in 4% paraformaldehyde, 4% sucrose, Tyrode's solution (136 mM NaCl, 2.5 mM KCl, 2 mM CaCl$_2$, 1.3 mM MgCl$_2$, 10 mM Na-HEPES, 10 mM D-glucose, pH 7.3) for 15 min, permeabilized for 5 min in 0.25% Triton X-100, Tyrode's solution, and then incubated in 5% Normal Donkey Serum (NDS), Tyrode's solution for 30 min at 37°C for blocking. Primary antibodies (anti-tdTomato, Abcam, chicken, 1:5,000; anti-MAP2, Synaptic Systems, guinea pig, 1:1,000; anti-tau, EMD Millipore, Mouse, 1:1,000 anti-PSD-95, Abcam, mouse, 1:1,000, anti-gephyrin, Synaptic Systems, mouse, 1:1,000) diluted in Tyrode's solution with 5% NDS were incubated for 2 h at 37° C. Then, appropriate secondary antibodies (Thermo Fisher Scientific, 1:1,000) diluted in Tyrode's solution with 5% NDS were incubated for 45 min at 37°C.

**Biochemical fractionation and immunoblot analysis**

Whole brains were homogenized by motorized tissue grinder in ice-cold homogenization buffer (0.32 M sucrose, 10 mM HEPES, 2 mM

EDTA, 2 mM EGTA containing protease and phosphatase inhibitors). Subcellular fractionation of mouse brains was performed as described (Wyszynski *et al*, 1998). Briefly, the homogenates were centrifuged at 900 *g* for 10 min (the resulting pellets are P1). The resulting supernatants were centrifuged again at 12,000 *g* for 15 min. The pellets were resuspended in homogenization buffer and centrifuged at 13,000 *g* for 15 min (the resulting pellets are P2 or crude synaptosomes).

The postsynaptic density (PSD) fraction was acquired as previously published (Bermejo *et al*, 2014). In brief, a half (by volume) of the resuspended P2 sample was homogenized and lysed via hypoosmotic shock for 30 min. This was followed by ultracentrifugation at 25,000 *g* for 20 min. The pellet was resuspended with a 0.32 M HEPES-buffered sucrose solution and gently placed at the top of a 3-part (1.2, 1.0, 0.8 M HEPES-buffered sucrose solution) discontinuous sucrose gradient and ultracentrifuged at 150,000 *g* for 2 h. The synaptic plasma membrane layer (at the interface of 1.2 M and 1.0 M layers) was extracted and ultracentrifuged in a 0.32 M HEPES-buffered sucrose solution at 200,000 *g* for 30 min. The pellet was resuspended in a buffer solution (50 mM HEPES, 2 mM EDTA), and 50 µl of the resuspended pellet was set aside as the synaptic plasma membrane (SPM) fraction. The resuspended pellet was added to a detergent solution (0.5% Triton X-100, 50 mM HEPES, 2 mM EDTA) and rotated for 15 min, after which the sample was ultracentrifuged at 32,000 g for 20 min. The pellet was again resuspended in the detergent solution and ultracentrifuged at 32,000 *g* for 20 min. The resulting pellet was resuspended in 50 µl of a 50 mM HEPES buffered and 2 mM EDTA solution and used as the postsynaptic density (PSD) sample. The detergent and ultracentrifugation procedures were performed twice to ensure to exclude presynaptic samples from PSDs (PSD II). All solutions mentioned were buffered at pH 7.4 (except for the hypoosmotic shock, which was done first in non-buffered double-distilled water followed by quick pH adjustment after homogenization), containing protease and phosphatase inhibitors. All procedures were performed at 4°C.

For micro-dissection of SR and SLM samples, dorsal hippocampal CA1 sections (400 µm) were sliced in sCSF with a vibratome (VT1200s, Leica). Under light microscopy, SR and SLM regions, which are readily discernible by their different darkness (SLM is darker), were manually dissected on an ice-cold platform, and four animals were pooled for one sample. Each sample was homogenized in 60–80 µl of ice-cold homogenization buffer, and the whole lysates were used without any further fractionation. A total of 32 of juvenile (3 weeks) male mice were used to make *n* number of 8 pairs by pooling 4 pairs of mice. The band was analyzed using Odyssey imaging program.

Corpus callosum samples were prepared as described previously (Li *et al*, 2016). Briefly, coronal brain sections (400 µm) containing the corpus callosum were sliced in aCSF using a vibratome (VT1200s, Leica). The corpus callosum was dissected under light microscope, and 3–4 slices were used per mice. For control, whole coronal sections containing corpus callosum (without any dissection) were used, for which a single slice per mice was used.

The following primary antibodies were used in immunoblotting analysis. Antibodies against PTPδ C-term (#2063, RPAMVQTED-QYQFCYRAALEYLGSFDHYAT), PTPδ N-term (#2061, IIQHKPKN-SEEPYKEIDGIATTRYSVAGLSPYSDYEFR), and PTPσ (#2135,

EEPPRFIREPKDQI) were generated using the indicated peptides, respectively. The following antibodies were previously described: NGL-1 (#2040) (Um *et al*, 2018), NGL-3 (#1948) (Lee *et al*, 2014), GluA1(#1193), GluA2 (#1195) (Kim *et al*, 2009), liprin-α (#1289) (Ko *et al*, 2003), SynGAP1 (#1682) (Kim *et al*, 2009).The following antibodies were commercially purchased: GluN2A (Alomone AGC-003), GluN2B (Neuromab 75-101), IL1RAPL1 (Proteintech, 21609-1-AP), IL1RAcP (Millipore, ABT333), Slitrk2 (Abcam ab67305), Slitrk3 (Abcam ab67306), mCherry (Abcam, ab125096), pTyr (4G10) (Millipore, 05-321), HCN1 (Neuromab 75-110), Bassoon (Stressgene, VAM-PS003), p-Src (Cell Signaling, 2105), PSD-95 (Neuromab 75-028), α-tubulin (Sigma T5168), β-actin (Sigma A5316), CaMKIIα/β (Cell Signaling 3362).

## Coimmunoprecipitation

For co-IP, transfected HEK293T cells or mouse brains were lysed with lysis buffer (1% NP-40, 50 mM Tris–HCl (pH8.0), aprotinin, benzamidine, leupeptin, pepstatinx1000, PMSF). The lysates were incubated with primary antibodies overnight at 4°C. Then, the samples were incubated with magnetic beads (Bio-Rad, Sure-BeadsTM, Cat#161-4023) for 2 h at 4°C. Samples were thoroughly washed three times with washing buffer (1×PBS, 0.1% NP-40, all protease inhibitors) and after eluted in 2×SDS.

## Brain slices for electrophysiology

Acute sagittal brain slices were obtained by anesthetizing P21–28 mice with isoflurane (Terrell) and extracting the brain into a 0°C dissection buffer containing, in mM: 212 sucrose, 25 NaHCO$_3$, 5 KCl, 1.25 NaH$_2$PO$_4$, 10 D-glucose, 2 sodium pyruvate, 1.2 sodium ascorbate, 3.5 MgCl$_2$, 0.5 CaCl$_2$, and bubbled with 95% O$_2$/5% CO$_2$. Dorsal hippocampal slices (300 µm) generated using a vibratome were transferred to a 32°C holding chamber containing a solution of artificial cerebrospinal fluid (aCSF; in mM: 125 NaCl, 25 NaHCO$_3$, 2.5 KCl, 1.25 NaH$_2$PO$_4$, 10 D-glucose, 1.3 MgCl$_2$, 2.5 CaCl$_2$). Slices were recovered at 32°C for 30 min and at room temperature (20–25°C) for 30 min. Then, slices were transferred to a recording chamber, where all electrophysiological experiments were performed at 28°C with circulating aCSF. Cells were visualized under differential interference contrast illumination in an upright microscope (B50WI, Olympus).

## Whole-cell recording

For whole-cell voltage-clamp recordings, thin-walled borosilicate capillaries (30-0065, Harvard Apparatus) were used to make pipettes with resistance 2.3–3.5 MΩ via a two-step vertical puller (PC-10, Narishige). For sEPSC and mEPSC recordings, pipettes were filled with an internal solution composed of, in mM, 117 CsMeSO$_4$, 10 EGTA, 8 NaCl, 10 TEACl, 10 HEPES, 4 Mg-ATP, 0.3 Na-GTP, and 5 QX-314. For mIPSC recordings, the internal solution contained, in mM, 115 CsCl, 10 EGTA, 8 NaCl, 10 TEACl, 10 HEPES, 4 Mg-ATP, 0.3 Na-GTP, and 5 QX-314. For cell property experiments, pipettes were filled with an internal solution containing, in mM, 137 K-gluconate, 5 KCl, 10 HEPES, 0.2 EGTA, 10 Na$_2$-phosphocreatine, 4 mM Mg-ATP, and 0.5 Na-GTP. All internal solutions were titrated to pH 7.35 and adjusted to the osmolarity of 285 mOsm. For sEPSC

experiments, 60 μM picrotoxin (Sigma) was added to the aCSF. For mEPSC experiments, 60 μM picrotoxin and 0.5 μM tetrodotoxin (Tocris) were added. For mIPSC experiments, 10 μM NBQX (Tocris), 50 μM D-AP5 (Tocris), and 0.5 μM tetrodotoxin (Tocris) were added. Access resistance was maintained as to be no > 20 MΩ or else excluded from data acquisition. Signals were filtered at 2 kHz and digitized at 10 kHz under control of the Multiclamp 700B Amplifier (Molecular Devices) and the Digidata 1550 Digitizer (Molecular Devices). Cells were approached with the internal solution-filled pipette to make giga seal, after which cells were gently ruptured via suction and maintained thereafter at −70 mV. After voltage-clamped cells were stabilized (~3 min post-rupture), recordings were obtained. Access resistance was monitored throughout the stabilization period and immediately before and after the data acquisition. The acquired data were analyzed using Clampfit 10 (Molecular Devices).

## Field recording

For field EPSP (fEPSP) recordings, baseline responses were collected at 0.07 Hz with a stimulation intensity that yielded a half-maximal response. Once a stable baseline response was acquired, excitatory transmissions were evoked at a set series of increasing stimuli using an isolated pulse stimulator (A-M Systems). fEPSP slopes and fiber volleys were then interpolated via linear fits to obtain an input/output model of basal evoked excitatory transmissions. For paired-pulse facilitation, pairs peak amplitude responses were obtained at indicated inter-pulse intervals and divided to obtain the PPF for each inter-pulse interval. Signals were filtered at 2 kHz and digitized at 1 kHz under control of the Multiclamp 700B Amplifier and the Digidata 1550 Digitizer. The acquired data were analyzed using Clampfit 10.

## PhosphoScan and SynGO analysis

PhosphoScan proteomics of WT and *Ptprd*-knockout brain samples were obtained via the PTMScan Technology services offered by Cell Signaling Technology. In brief, whole-brain samples were obtained by anesthetizing in isoflurane and extracting the brain followed by freezing in liquid nitrogen. Samples were first homogenized and protease-digested, after which lyophilized peptides obtained by loading directly onto a 50 cm × 100 μm PicoFrit (New Objective) capillary column packed with C18 reversed-phase resin. Phosphotyrosine pY-1000 Motif Antibody (#8803, Cell Signaling Technology) was used to immobilize the target motif peptides followed by immunoprecipitation. Peptides were washed, eluted, and analyzed by LC-MS/MS. MS/MS spectra were evaluated using the SEQUEST and the Core platform from Harvard University. Assignment and relative quantification of the sequences to the MS/MS spectra was done with the Sorcerer program (Sage-N). Searches were performed against the most recent update of the Uniprot *Mus musculus* database with mass accuracy of ± 50 ppm for precursor ions and 0.02 Da for product ions. Results were filtered with mass accuracy of ± 5 ppm on precursor ions and the presence of the intended motif.

For SynGO analysis, significantly changed motifs ($P < 0.05$) were extracted, and the gene names corresponding to the motifs (see Dataset EV1, "SynGO Input" tab) were analyzed using the SynGO evidence-based resource for annotation of synaptic proteins (https://www.syngoportal.org/). Enrichment data for each ontology term containing at least 3 genes from the input list were obtained by using a one-sided Fisher exact test and comparison with all brain expression genes as the background set. *P*-values were adjusted for multiple comparisons using the false discovery rate approach. Data are visualized using sunburst graphs of localization (GO, cellular component) and function (GO, biological process). The sunburst graphs depict parent and child ontology terms of the synapse in concentric rings with the more specific terms outward. See the interactive Ontology section on the home page of https://www.syngoportal.org.

## Behavioral tests

All behavioral assays were performed using age-matched mice (2- to 5-month-old mice) during light-off periods (active phase). Before the tests, mice were handled at least for 3 days (10 min/day).

### LABORAS test

The Laboratory Animal Behavior Observation Registration and Analysis System (LABORAS™, Metris) test was used to measure continuous mouse movements in an isolated environment without researcher's presence for 72 h. Mice in their home cages without prior isolation were individually caged atop a vibration-sensitive platform inside a sound-attenuated room. We did not try prior isolation in home cages because the first 1 or 2 days in the Laboras cage likely function as an isolation condition. We also measured the weight of water, chow, and individual subject before and after the recording. Data acquisition and analysis were performed using LABORAS 2.6 program (Metris).

### Open-field test

Subject mice were placed in a white acryl open-field box (40 × 40 × 40 cm), and the movements were recorded for 60 min for adult mice. Illuminations were set at 0 lux. The distance moved and time spent in the center arena (20 × 20 cm) were measured using EthoVision XT 10.1 program (Noldus).

### Elevated plus-maze test

Subject mice were placed in the center region of an elevated plus-arm maze, with two open and closed (30-cm walled) arms (5 × 30 cm each). The maze was situated 50 cm above the floor. Mouse movements were recorded for 10 min. The open arm was illuminated at ~200 lux. Time spent in each set of arms and the total distance moved were measured using EthoVision XT 10.1 program (Noldus).

### Light–dark test

The light/dark box consisted of an open roof white ("light") chamber conjoined to a closed black ("dark") chamber with a small entrance to allow free movements between the two chambers. The light chamber was illuminated at 200 lux. Entry to the light chamber was counted only when the entire body of the subject crossed the

entrance. Time spent in each chamber and the number of entries were measured using EthoVision XT 10.1 program (Noldus).

## Electroencephalography and electromyography

Custom probes were assembled using the following materials: 6-position array rectangular housing connector 2 mm (DF11-6DS-2C, Hirose Electric Co.), 30 AWG socket contact gold crimp (DF11-30SCFA, Hirose Electric Co.), stranded stainless steel hookup wire (AS633, Cooner Wire), and M1 $1 \times 3$ mm microscrews (M1, bolt24.-com). In brief, 4 strands of wire 25 mm long were cut and 5 mm of each end stripped. One end was soldered to a crimp and the other to a screw head. 2 strands were cut at cut 27 mm length, 5 mm of each end stripped, one end soldered to a crimp, and the other end rolled into a ball. The assembled probe was installed via stereotaxic surgery. Mice were first anesthetized with a ketamine (50 mg/ml, Yuhan) and xylazine (5.5 mg/ml, X1126, Sigma-Aldrich) cocktail mixed in 1:1 ratio and injected intraperitoneally (IP) at 5 ml/kg. After mice were confirmed anesthetized, they were affixed onto the stereotaxic platform (Model 940, Kopf Instruments) and leveled at the bregma-lambda axis. Screws of the probe were securely screwed onto the skull at the following [medial–lateral (ML), anterior–posterior (AP)] coordinates, in mm with respect to bregma, for electroencephalography (EEG) signal acquisition: Channel 1 (Ch1; +1.5, +1.5), Channel 2 (Ch2; +3.0, −3.5), EEG reference (Ref$_E$; +0, −6.5), and Animal ground (Gr; −3.0, −3.5). For electromyography (EMG) signal acquisition, the two strands with wire balls were each inserted deep into the left (EMG reference, Ref$_M$) and right (EMG) trapezius muscles. For recording, the mice were placed in a white acryl box ($25 \times 25 \times 35$ cm), where the signals were fed via a slipring (WR-M006, woorisr.com) to a high-density electrophysiology acquisition system (hardware: Digital Lynx 4SX; software: Cheetah 5.0; NeuraLynx) with synchronized video capture. Mice were kept in the recording system for 48 h, of which the last 24 h were used as data. The analysis was done using a combination of the SleepSign (Kissei Comtec Co.) software and custom-written Matlab code. The 24-h data were parsed into 4-s epochs and the state of the mice (WAKE, NREM, or REM) determined by a combination of EEG fast Fourier transform analysis, EMG intensity analysis, and video confirmation (where required). Ch1 and Ch2 were not combined to result in a single EEG signal, but rather, each channel was taken as a separate signal for consideration and analysis.

## Experimental design, data acquisition, analysis, and statistics

All experiments were done in a strictly double-blind manner. Only male mice were used for the experiments and grouped solely based on their genotype. Numbers and detailed statistical analysis of all the results reported are organized by figures in Dataset EV2. Statistical analyses were justified and the data checked to see whether it meets the assumptions of the statistical tests employed. For two-way ANOVA and two-way repeated-measures (RM) ANOVA, normality was assumed but not formally tested. Sample sizes were determined based on the relevant published literature (Shah, 2011) and/or by the nature of the experimental design. Statistical tests were performed using SigmaPlot 12.0 (Systat Software Inc.). Outliers were excluded based on the results of the ROUT test ($Q = 1\%$). Post-hoc tests

for ANOVAs were done only if either the interaction or both main factors were significant.

**Expanded View** for this article is available online.

## Acknowledgements

This work was supported by the National Research Foundation of Korea (NRF-2017R1A5A2015391 to Y.B) and the Institute for Basic Science (IBS-R002-D1 to E.K). We would like to especially thank Drs. Tae Kim (Gwangju Institute of Science and Technology), Jeonghoon Woo (Chungnam Techno Park), and Eunee Lee (Yonsei University) for their invaluable inputs during the EEG recordings and analysis, as well as to Dr. Ganghoo Kim (Samsung Electronics) for his advice on numerous statistical issues and analyses.

## Author contributions

HR, YC, SK, and EY performed tdTomato and other imaging experiments; HJ, YC, and SK immunoblot experiments; SL performed neuron culture experiments; HH performed EM experiments; HR and YC performed electrophysiological experiments; HR performed tyrosine phosphorylation analysis; FK performed SynGO analysis; HR and SK performed PSD fractionation; HR, YC, HK, SK, and WS performed behavioral experiments; HR and YC performed EEG experiments and analysis; HK, ABS, MV, YCB, and EK designed research and wrote the manuscript.

## Conflict of interest

The authors declare that they have no conflict of interest.

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
