## [Review Process File · The EMBO Journal]

Splice-dependent trans-synaptic PTP δ -IL1RAPL1 interaction regulates synapse formation and NREM sleep

Haram Park, Yeonsoo Choi, Hwajin Jung, Seoyeong Kim, Suho Lee, Hyemin Han, Hanseul Kweon, Suwon Kang, Woong Seob Sim, Frank Koopmans, Esther Yang, Hyun Kim, August B. Smit, Yong Chul Bae, Eunjoon Kim

Review timeline:

Submission date:	30 November 2019
Editorial Decision:	23 December 2019
Revision received:	24 February 2020
Editorial Decision:	13 March 2020
Revision received:	17 March 2020
Accepted:	23 March 2020

Editor: Karin Dumstei

Transaction Report:

1st Editorial Decision

23 December 2019

Thanks for submitting your manuscript to The EMBO Journal. Your study has now been seen by three referees and their comments are provided below.

As you can see from the comments, the referees find the analysis interesting and suitable for publication here. The referees bring up a number of points that should be further addressed. Most of them should be fairly straightforward to resolve. Some of the points raised by referee #1 might be a bit more tricky - I am happy to discuss them further if needed. I would like to keep the behavioral data in the MS (referee #1 point #10).

Thank you for the opportunity to consider your work for publication. I look forward to your revision.

REFeree REPORTS

Referee #1:

Park et al. present a very interesting study in the field of synaptogenesis research. The general focus is on PTPd, a member of the family of LAR receptor protein tyrosine phosphatases (LAR-RPTPs) and a known player in synaptogenesis. In this context, PTPd is particularly interesting because it is thought to operate via multiple trans-synaptic interaction partners and because some of these interactions are controlled by alternative splicing of PTPd mRNAs. Further, PTPd is implicated in multiple pathophysiological conditions.

The work is very substantial and built on three (!) new mutant mouse lines, a knock-in mutant PTPd reporter line (tdTomato-tag), a new knock-out line, and a line carrying a deletion of a small exon (miniexonA) that is known to be required for the interaction of PTPd with its cognate interaction partner IL1RAPL1. The 'gist' of the study is that PTPd controls the formation of specific excitatory synapses in the hippocampus via its interaction with IL1RAPL1 and that this depends on the miniexonA. Further data show that knock-outs of PTPd or miniexonA result in hyperactivity and sleep aberrations.

In essence, the study is of EMBO-Journal caliber - but because it covers a lot of ground and employs a substantial set of new mouse genetics tools, there remain a few areas where more information is required, as far as I see it.

I am commenting in the order in which the respective issues appear in the manuscript:

1. In order to stress the uniqueness of their study, the authors introduce it in a way that borders on belittling previously published work. One corresponding issue relates to the role of alternative splicing in synapse development, where the text states at one point that "it remains unclear whether alternative splicing regulates synapse development in vivo". Unless phrased in a much more differentiated manner (I assume the authors are specifically alluding to synapse density), such a statement is essentially wrong. The same is true for the subsequent statement that it is unknown whether alternative splicing of synaptic adhesion proteins regulates specific brain function. Here, several studies on Neurexins show that their alternative splicing affects synapses in multiple circuits and document this at a level of functional analysis that goes substantially beyond the analysis depth of the present paper. I suggest to phrase all related passages in the introduction much more carefully in order to reflect the published literature in a more balanced manner. I personally think that these statements may not even be necessary at all, and neither is the rather major emphasis on the miniexonA splicing. At least as important, as far as I see it, is the insight the study can provide into the in vivo function of this fascinating LAR-RPTP - e.g. expression patterns, subcellular localizations, synapse specificity, role in synapse development and function, role of miniexonA splicing, or interaction targets.

2. In the introduction already, the authors describe their new mouse lines as "transgenic". I know that there is confusion in the use of this term, but my understanding is that "transgenic" implies the transfer of one gene to a (random) locus in another genome. The authors did much better as they actually mutated the endogenous PTPd locus by homologous recombination (ES cells or CRISPR-HDR). Why not use the more appropriate terms "knock-in" and "knock-out"?

3. The PTPd-tdTomato knock-in is a great idea given that endogenous PTPd is difficult to detect in cells and tissues with currently available antibodies, but I think more validation of the new mouse tool is required before one can take the data it reports at face value. After all, the tag is substantial in size and there is the risk that it affects trafficking, subcellular localization, or even interaction with substrates. I understand that the lack of useful antibodies prevents a comparison of the distributions of WT and tagged PTPd. Or is there an antibody to PTPd that could be used to validate the localization of PTPd-tdTomato in the knock-in - maybe at least in cultured neurons? If not, the next best thing would be to show with a careful subcellular fractionation that the two variants co-segregate across all relevant subcellular fractions (synaptic and non-synaptic). A second test that could be done is to see whether cellular/synaptic pTyr patterns are the same in WT and PTPd-tdTomato knock-in (analogous to Fig. 3, for instance). This is also critical in view of the fact that

PTPd-tdTomato seems to be expressed at much lower levels than the WT variant (a phenomenon that I would like to see quantified). Ultimately, one should also seriously consider testing the PTPd-tdTomato knock-in for proper synapse function (e.g. in SLM). Another issue that needs some consideration and discussion in my view is the apparently very high expression level of PTPd-tdTomato in the corpus callosum (and the anterior commissure). This is weird for a synaptic protein. Is this real and really within the axons, and can this be validated by Western blotting?

4. Fig. 1d is fascinating but also confusing. It shows hippocampal primary neurons from PTPd-tdTomato knock-in hippocampus and demonstrates the axonal localization of PTPd-tdTomato. What strikes me, though, is the fact that the cells that are cultured (i.e. hippocampal pyramidal and granule cells) seem to be among the weakest PTPd-tdTomato expressors in the entire brain (see Fig. 1c - where the synaptic neuropil containing presynapses of hippocampal pyramidal and granule cells is essentially blank). Why did the authors not use cortical primary neurons for this type of analysis as the cortex seems to be full of PTPd-tdTomato? I note that the authors show in Fig. S1 analogous data from "hippocampal/cortical" neurons, but it is unclear to me what these are and what proportion of the cells are cortical. A final issue regarding Fig. 1d is the strong perisomatic PTPd-tdTomato labeling in the bottom panel series. Are these inhibitory (pre)synapses containing PTPd-tdTomato?

5. The question just posed above leads me to one key conclusion of the present paper, which is that PTPd is specifically present at excitatory presynapses in the cells/tissues tested. Essentially, I am not yet convinced that this is properly demonstrated. I like EM and synapse-type specificity of a given target can be studied with EM. However, this requires careful experimentation and quantitative analyses with proper statistics. Fig. 1e-1g is presented to demonstrate a specific localization of PTPd-tdTomato at excitatory presynaptic terminals in the SLM. However, the corresponding large field images (Fig. S1f-S1h) show abundant DAB signals in areas that I at least cannot attribute to excitatory presynaptic terminals. I think this EM analysis needs to be done quantitatively with statistics (e.g. synaptic, outside synapses, percentage of excitatory synapses labeled, etc.). In conjunction with my concerns outlined in my point 4, I was wondering why the authors did not try to address the issue of synapse-type specificity using high-resolution or super-resolution fluorescence microscopy - to study synapses first in cultured cells and then in the corresponding tissue (note that, in contrast, Fig. 1d and Fig. 1e-1g deal with different synapses). In this manner, a much larger dataset on a much larger variety of brain regions and synapse types could be generated, probably with less effort than by using EM.

6. The specific deficit in SLM excitatory transmission in the PTPd knock-out (Fig. 2i) is a nice indirect validation of the localization data obtained with the PTPd-tdTomato knock-in. The authors associate this with a 40% reduction in PSD number per area in the SLM as assessed by EM. A critical open question here is whether presynaptic terminals are also reduced in number. Are these synapses really gone or just dysfunctional? Or are just the PSDs gone? Immunofluorescence labeling for pre- and postsynaptic components should allow to resolve this issue.

7. What struck me in view of the data shown in Fig. 2l was the fact that the authors could not detect a change in PSD95 levels (or in the levels of any other marker of excitatory postsynapses) in micro-dissected SLM from PTPd knock-outs (Fig. S3). This is totally unexpected in view of the fact that 40% of PSDs are gone (Fig. 2l). Is the micro-dissection possibly flawed?

8. The authors then moved on directly to pTyr proteomics of whole brains. In view of the data provided up to this point, this feels like a huge step. I concede that it yielded a very important finding, i.e. that the PTPd interaction partner IL1RAPL1 is less Tyr-phosphorylated in the PTPd knock-out. But was any quantitative proteomics done on synaptic fractions to see protein composition correlates of the 40% PSD loss seen in SLM? I am just wondering - as it seems so obvious - and would not want to send the authors back to the bench for this at this juncture. And: Are overall levels of IL1RAPL1 altered anywhere in the brain of the PTPd knock-out (the second WT-knock-out pair in Fig. 4a looks a bit as if IL1RAPL1 levels are lower in the knock-out).

9. The recapitulation of the PTPd knock-out phenotype by the knock-out of *miniexonA* is nice and an essential addition to the story. A question that comes to mind when looking at Fig. 4h and 4i is whether IL1RAPL1 is at these postsynapses. The authors state in their title that "trans-synaptic PTPd-IL1RAPL1 interaction regulates synapse formation", which is probably true but only inferred based on previous data and the assumed effect of the genetic tools used. It seems important to show

at least that IL1RAPL1 is postsynaptic in the SLM (showing that the biochemical changes happen there will be more or less impossible).

10. The behavioral data are intriguing but somewhat detached from the rest of the paper as Fig. 1-4 deal with regions of the brain that are probably not directly involved in the behavioral defects of the mutant mice. The discussion of these data reflects this detachment. I am not fully convinced that they need to be part of this study.

Conclusion:

I would really like to see this study published in EMBO Journal. But I think - particularly for EMBO Journal - a somewhat more elaborate analysis of the cell biology, which is at the core of the study, is warranted. In essence, even some of the key conclusions are not watertight yet, e.g. the synapse-type specificity of PTPdelta. I think resolving the issues I outlined above is necessary, but I am willing to discuss this with the authors.

Referee #2:

PTPs are thought to mediate adhesion between pre-and postsynaptic site but many details remain to be determined. Here Eunjoon Kim and co-workers show that PTPdelta has its highest expression level in the SML layer of the neuropil in the hippocampus with remarkably little of this protein present in neighboring layers including SR. Immuno-EM analysis indicates a mostly presynaptic localization of PTPdelta. KO of PTPdelta as well as elimination of the small meA insert that mediates binding to the postsynaptic adhesion protein IL1RAPL1 lead to reduction in the strength of synapses in SML but not SR or SO of the hippocampal CA1 area. In parallel, synapse number but not size is reduced in the SML of these mice. The authors identified IL1RAPL1 as the relevant interaction partner by an unbiased proteomic screen for changes in tyrosine phosphorylation, another strength of this manuscript. In this screen IL1RAPL1 showed by a large margin the highest change in tyrosine phosphorylation. According to these functional studies, PTPdelta and especially its bind to IL1RAPL1 has a selective role in mediating synapse formation in a define hippocampal layer.

Furthermore, full PTPdelta KO as well as conditional KO in glutamatergic but not so much GABAergic neurons and the meA deletion all result in hyperactivity as tested in open field and in Laboras cages. Finally, EEG and EMG recordings from the meA deleted and the conditional PTPdelta KO in glutamatergic neurons showed a reduction in nonREM sleep and delta power.

The experiments are very thorough and rigorously executed and conclusions as clearly supported and of high interest. I only have a modest concern

Modest Concern:

Fig. 2: that PTPdelta immunolabeling is largely not overlapping with MAP2B labeling is not truly indicative of the labeling being not within the dendrite. Microtubules and its associated MAP2B are in the center of dendrites and proteins in or near the dendritic plasma membrane typically do not overlap with MAP2B. However, this concern is alleviated by the immuno-EM analysis, which shows a mostly presynaptic localization of PTPdelta at least at the synapse.

Minor Concern:

The various CAM should be better defined. For instance, for somebody outside the field it is not clear what NGL3 stands. The authors might want to consider better defining all CAMs with full names in the introduction or provide a table with full names.

Referee #3:

In this paper Park et al. studied extensively the function of the receptor tyrosine phosphatase PTPdelta. Using extensively mouse genetics they clearly showed that PTPdelta in mature neurons is essentially present only at the excitatory presynaptic membrane. The total deletion of PTPdelta in

mice causes a specific decrease in excitatory synapse development and strength. Interestingly this is mediated by tyrosine dephosphorylation and synaptic loss of IL1RAPL1, a major postsynaptic partner. This was elegantly demonstrated by accurate phosphoproteomic analysis and using a PTP δ -mutant mouse lacking the PTP δ -meA insert, thus expressing a PTPdelta mutant lacking the ability to interact only with IL1RAPL1 among the other postsynaptic partners. Finally, they also provide strong evidence that mice deleted of PTPdelta exhibit abnormal sleep behavior and non-REM rhythms suggesting that specific alternative splicing in PTPdelta and trans-synaptic adhesion with IL1RAPL1 regulates excitatory synapse development and sleep behavior. This is a very impressive and comprehensive paper that strongly demonstrates the synaptic function of PTPdelta. The data are very convincing and of top quality. The following two points should be addressed to clarify a couple of open questions.

1) The paper provides very nice and convincing data indicating that PTPdelta is a presynaptic protein localized at the excitatory synapses. They also showed, by EM analysis, that in the PTPdelta KO mice there is a reduction of PSD density in the SLM layer of the CA1, but no changes in the structure of the PSD. Considering that PTPdelta is a presynaptic phosphatase it should be important to analyze the structure of the presynapses by measuring for example size of the terminals and the density of synaptic vesicles.

2) To further strengthen the finding that PTPdelta deletion specifically reduces the localization of IL1RAPL1 to PSD, it will be interesting to see if some of the other PTPdelta interacting postsynaptic CAMs remain localized at PSD in PTPdelta and PTPdelta-meA KO mice. If antibodies are available the authors should analyze the biochemical distribution on PSD fraction of some of the other postsynaptic CAMs interacting with PTPdelta.

Point-by-point responses to referees' comments (EMBOJ-2019-104150)

Referee #1:

Park et al. present a very interesting study in the field of synaptogenesis research. The general focus is on PTPd, a member of the family of LAR receptor protein tyrosine phosphatases (LAR-RPTPs) and a known player in synaptogenesis. In this context, PTPd is particularly interesting because it is thought to operate via multiple trans-synaptic interaction partners and because some of these interactions are controlled by alternative splicing of PTPd mRNAs. Further, PTPd is implicated in multiple pathophysiological conditions.

The work is very substantial and built on three (!) new mutant mouse lines, a knock-in mutant PTPd reporter line (tdTomato-tag), a new knock-out line, and a line carrying a deletion of a small exon (miniexonA) that is known to be required for the interaction of PTPd with its cognate interaction partner IL1RAPL1. The 'gist' of the study is that PTPd controls the formation of specific excitatory synapses in the hippocampus via its interaction with IL1RAPL1 and that this depends on the miniexonA. Further data show that knock-outs of PTPd or miniexonA result in hyperactivity and sleep aberrations.

In essence, the study is of EMBO-Journal caliber - but because it covers a lot of ground and employs a substantial set of new mouse genetics tools, there remain a few areas where more information is required, as far as I see it.

→ We appreciate the encouraging comments of the reviewer.

I am commenting in the order in which the respective issues appear in the manuscript:

1. *In order to stress the uniqueness of their study, the authors introduce it in a way that borders on belittling previously published work. One corresponding issue relates to the role of alternative splicing in synapse development, where the text states at one point that "it remains unclear whether alternative splicing regulates synapse development in vivo". Unless phrased in a much more differentiated manner (I assume the authors are specifically alluding to synapse density), such a statement is essentially wrong. The same is true for the subsequent statement that it is unknown whether alternative splicing of synaptic adhesion proteins regulates specific brain function. Here, several studies on Neurexins show that their alternative splicing affects synapses in multiple circuits and document this at a level of functional analysis that goes substantially beyond the analysis depth of the present paper. I suggest to phrase all related passages in the introduction much more carefully in order to reflect the published literature in a more balanced manner. I personally think that these statements may not even be necessary at all, and neither is the rather major emphasis on the miniexonA splicing. At least as important, as far as I see it, is the insight the study can provide into the in vivo function of this fascinating LAR-RPTP - e.g. expression patterns, subcellular localizations, synapse specificity, role in synapse development and function, role of miniexonA splicing, or interaction targets.*

→ We appreciate the thoughtful comments and accordingly modified the text in the Introduction to more carefully include previous studies on alternative splicing-dependent regulation of synapse development and brain functions. In addition, we modified the text in a way to give more weights on the in vivo functions of LAR-RPTPs rather than alternative splicing-dependent 'synapse development'; i.e. we removed the two sentences that state that whether alternative splicing regulates synapse development or brain function is unclear.

2. In the introduction already, the authors describe their new mouse lines as "transgenic". I know that there is confusion in the use of this term, but my understanding is that "transgenic" implies the transfer of one gene to a (random) locus in another genome. The authors did much better as they actually mutated the endogenous PTPd locus by homologous recombination (ES cells or CRISPR-HDR). Why not use the more appropriate terms "knock-in" and "knock-out"?

→ We now use "knock-in" and "knock-out" terms in the Introduction.

3. The PTPd-tdTomato knock-in is a great idea given that endogenous PTPd is difficult to detect in cells and tissues with currently available antibodies, but I think more validation of the new mouse tool is required before one can take the data it reports at face value. After all, the tag is substantial in size and there is the risk that it affects trafficking, subcellular localization, or even interaction with substrates. I understand that the lack of useful antibodies prevents a comparison of the distributions of WT and tagged PTPd. Or is there an antibody to PTPd that could be used to validate the localization of PTPd-tdTomato in the knock-in - maybe at least in cultured neurons?

→ We agree with the reviewer that there is the possibility that the C-terminal tdTomato tag may interfere with protein trafficking, localization, and protein interaction. Unfortunately, however, we do not have PTP δ antibodies that can be used to visualize endogenous PTP δ proteins and to compare the synaptic localization of endogenous PTP δ and PTP δ -tdTomato proteins. However, we now show the data from cultured neurons demonstrating that PTP δ -tdTomato is mainly targeted to excitatory synapses relative to inhibitory synapses (see our response to your review comment #5 below). In addition, our new quantitative EM analysis indicates that PTP δ -tdTomato proteins are mainly at excitatory presynaptic sites relative to inhibitory presynaptic sites (85% vs. 10%; see our response to your review comment #5 below).

If not, the next best thing would be to show with a careful subcellular fractionation that the two variants co-segregate across all relevant subcellular fractions (synaptic and non-synaptic).

→ We performed the requested experiment and found endogenous PTP δ and PTP δ -tdTomato proteins similarly distribute to different subcellular brain fractions (P1, S2, P2, S3, P3, LP1, LS1, LP2 fractions) (**Fig EV1J**).

A second test that could be done is to see whether cellular/synaptic pTyr patterns are the same in WT and PTPδ-tdTomato knock-in (analogous to Fig. 3, for instance). This is also critical in view of the fact that PTPδ-tdTomato seems to be expressed at much lower levels than the WT variant (a phenomenon that I would like to see quantified). Ultimately, one should also seriously consider testing the PTPδ-tdTomato knock-in for proper synapse function (e.g. in SLM).

→ We appreciate this thoughtful comment. In our opinion, pTyr pattern, or synaptic transmission in SLM, is an indirect measure for synaptic localization of PTPδ-tdTomato and unlikely to give us conclusive evidence. Regarding the reduced levels of PTPδ-tdTomato, we added the following new data and text to Results: “The reduced levels of PTPδ-tdTomato proteins relative to endogenous PTPδ proteins, both of which represent the C-terminal fragment after proteolytic cleavage in the middle of the protein (Chagnon, Uetani et al., 2004), could be attributable to the destabilization of the hybrid PTPδ-tdTomato protein. Alternatively, the tdTomato portion of the PTPδ-tdTomato protein may suppress the C-terminal PTPδ antibody to bind to its cognate antigen. Notably, the N-terminal PTPδ antibody detected similar amounts of the N-terminal fragment of PTPδ-tdTomato (after cleavage) (**Fig 1B**).”

Another issue that needs some consideration and discussion in my view is the apparently very high expression level of PTPδ-tdTomato in the corpus callosum (and the anterior commissure). This is weird for a synaptic protein. Is this real and really within the axons, and can this be validated by Western blotting?

→ We performed the suggested immunoblot analysis and found that PTPδ-tdTomato signals are indeed increased in the corpus callosum. We added the following text to Results: “In line with the previous suggestion for presynaptic functions of PTPδ, PTPδ-tdTomato signals were stronger in brain regions enriched with axonal fibers such as the corpus callosum, as supported by immunoblot analysis of corpus callosal samples (~50% higher in average intensity compared with the whole-brain sample, n = 4 mice, p = 0.0016, paired t-test).”

4. Fig. 1d is fascinating but also confusing. It shows hippocampal primary neurons from PTPδ-tdTomato knock-in hippocampus and demonstrates the axonal localization of PTPδ-tdTomato. What strikes me, though, is the fact that the cells that are cultured (i.e. hippocampal pyramidal and granule cells) seem to be among the weakest PTPδ-tdTomato expressors in the entire brain (see Fig. 1c - where the synaptic neuropil containing presynapses of hippocampal pyramidal and granule cells is essentially blank). Why did the authors not use cortical primary neurons for this type of analysis as the cortex seems to be full of PTPδ-tdTomato? I note that the authors show in Fig. S1 analogous data from "hippocampal/cortical" neurons, but it is unclear to me what these are and what proportion of the cells are cortical.

→ Our sincere apologies for our mistake in our description! We actually used mixed cultured neurons derived from two different brain regions (entorhinal cortex and hippocampus; 1:2 mixture) where hippocampal neurons could possibly act as postsynaptic neurons. We clarified this in the figure legends for both **Fig 1D** and **Fig EV1D,E**.

A final issue regarding Fig. 1d is the strong perisomatic PTPd-tdTomato labeling in the bottom panel series. Are these inhibitory (pre)synapses containing PTPd-tdTomato?

→ We think that the strong tdTomato signals likely represent abundant axonal fibers physically in contact with the cell body because we often observe strong signals for tau (axonal marker) around the cell body (see the image of tdTomato-tau colocalization below). In addition, these perisomatic tdTomato signals did not colocalize with gephyrin (inhibitory synaptic marker) (see the image of tdTomato-gephyrin colocalization below). Please find further details on PTPRD colocalization with PSD-95 but not with gephyrin in our answers to your comment #5 below and the new Fig 1E.

PTPRD-Tau localization

tdTomato-gephyrin colocalization

5. The question just posed above leads me to one key conclusion of the present paper, which is that PTPd is specifically present at excitatory presynapses in the cells/tissues tested. Essentially, I am not yet convinced that this is properly

demonstrated. I like EM and synapse-type specificity of a given target can be studied with EM. However, this requires careful experimentation and quantitative analyses with proper statistics. Fig. 1e-1g is presented to demonstrate a specific localization of PTP δ -tdTomato at excitatory presynaptic terminals in the SLM. However, the corresponding large field images (Fig. S1f-S1h) show abundant DAB signals in areas that I at least cannot attribute to excitatory presynaptic terminals. I think this EM analysis needs to be done quantitatively with statistics (e.g. synaptic, outside synapses, percentage of excitatory synapses labeled, etc.).

→ In response, we performed quantitative analysis of DAB signals in the EM images and found that ~85% of PTP δ -tdTomato signals are present in excitatory presynaptic terminals apposed to PSDs, ~10% are present in presynaptic terminals not apposed to PSDs, and ~5% are present in neuropils other than presynaptic terminals. (**Fig.11**)

In conjunction with my concerns outlined in my point 4, I was wondering why the authors did not try to address the issue of synapse-type specificity using high-resolution or super-resolution fluorescence microscopy - to study synapses first in cultured cells and then in the corresponding tissue (note that, in contrast, Fig. 1d and Fig. 1e-1g deal with different synapses). In this manner, a much larger dataset on a much larger variety of brain regions and synapse types could be generated, probably with less effort than by using EM.

→ We now show the data from cultured neurons (entorhinal cortex + hippocampus), which clearly show that the majority of the PTP δ -tdTomato signals are in contact with PSD-95 (excitatory synaptic marker) but not with gephyrin (inhibitory synaptic marker) (**Fig 1E**). Quantitative analysis indicated that PTP δ -tdTomato signals were more strongly (~2.7-folds) in contact with excitatory synapses (PSD-95) relative to inhibitory synapses (gephyrin) (**Fig EV1F**), in line with the strong (~85%) excitatory synaptic localization of PTP δ -tdTomato signals at excitatory synapses in the SLM region of the hippocampus (**Fig. 1F**). This difference (2.7:1 in cultured neurons and 85% vs 10% in EM) could be due to the difference between in vitro and in vivo contexts, or that the cultured neurons were from both cortical and hippocampal regions but not from the SLM region. This suggests the intriguing possibility that PTP δ may also be at excitatory synapses in addition to inhibitory synapses in other brain regions.

Addressing synapse-type specificity using high-resolution fluorescence microscopy and cultured neurons or tissues from various brain areas is indeed a very important task. We believe, however, that this question, which should be comprehensively and carefully addressed for convincing conclusions, constitutes a topic for a follow-up study. It is certainly possible that brain regions other than the hippocampal SLM may contain PTP δ proteins at inhibitory synapses in addition to excitatory synapses. We thus empathized this possibility briefly in Results as follows: “However, these results do not exclude the possibility that PTP δ proteins are present at inhibitory synapses in addition to excitatory synapses in brain regions other than the hippocampal SLM.”

6. The specific deficit in SLM excitatory transmission in the PTP δ knock-out (Fig. 2i) is a nice indirect validation of the localization data obtained with the PTP δ -tdTomato

knock-in. The authors associate this with a 40% reduction in PSD number per area in the SLM as assessed by EM. A critical open question here is whether presynaptic terminals are also reduced in number. Are these synapses really gone or just dysfunctional? Or are just the PSDs gone? Immunofluorescence labeling for pre- and postsynaptic components should allow to resolve this issue.

→ We performed a quantitative EM analysis and found that the density of presynaptic nerve terminals was decreased (~28%) in the *Ptprd*^{-/-} SLM, whereas the area of nerve terminals and the density of presynaptic vesicles were normal (**Fig. 2P–R**), suggesting that presynaptic structures are substantially eliminated together with postsynaptic structures. We appreciate the idea of immunofluorescence labeling but thought that parallel pre- and postsynaptic quantifications may be informative.

7. What struck me in view of the data shown in Fig. 2I was the fact that the authors could not detect a change in PSD95 levels (or in the levels of any other marker of excitatory postsynapses) in micro-dissected SLM from PTPd knock-outs (Fig. S3). This is totally unexpected in view of the fact that 40% of PSDs are gone (Fig. 2I). Is the micro-dissection possibly flawed?

→ We think that our micro-dissection procedure is reliable because SLM marker proteins (NGL-1 and HCN1) were enriched in our SLM samples. The discrepancy could be attributable to that the results in **Fig 2L** are from PSD samples, whereas those in **Fig. S3** (now **Fig EV3**) are from total lysates (now clarified in the figure legend; our sincere apologies!). These results suggest that synaptic proteins released from the eliminated excitatory synapses in mutant neurons may be present at certain non-synaptic sites without degradation. In support of this possibility, i.e. in **Fig. 4B and G**, levels of IL1RAPL1 proteins are decreased in PSDs but not decreased in the P2 fraction. We thus added the following text to Results: “The lack of changes in the levels of major postsynaptic proteins in the SLM region of *Ptprd*^{-/-} mice may be attributable to that total lysates, but not synapse-enriched preparations, were used for immunoblot analyses (see the results in **Fig 4** below).”

8. The authors then moved on directly to pTyr proteomics of whole brains. In view of the data provided up to this point, this feels like a huge step. I concede that it yielded a very important finding, i.e. that the PTPd interaction partner IL1RAPL1 is less Tyr-phosphorylated in the PTPd knock-out. But was any quantitative proteomics done on synaptic fractions to see protein composition correlates of the 40% PSD loss seen in SLM? I am just wondering - as it seems so obvious - and would not want to send the authors back to the bench for this at this juncture. And: Are overall levels of IL1RAPL1 altered anywhere in the brain of the PTPd knock-out (the second WT-knock-out pair in Fig. 4a looks a bit as if IL1RAPL1 levels are lower in the knock-out).

→ We have not performed a proteomic analysis of SPM (synaptic plasma membrane) or PSD samples because the amounts of proteins in PSD samples may not be sufficient for proteomic analysis, and SPM is a fraction where synaptic proteins would be minimally decreased in the mutant mice, as shown in **Fig. 4B and G**; we hope that the reviewer understands.

The decreasing tendency in IL1RAPL1 levels in the second WT-KO pair in

Fig. 4A is because a relatively small amount of sample was loaded, as shown by the β -actin immunoblot. We replaced the images in the figure with better ones.

9. The recapitulation of the PTPd knock-out phenotype by the knock-out of miniexonA is nice and an essential addition to the story. A question that comes to mind when looking at Fig. 4h and 4i is whether IL1RAPL1 is at these postsynapses. The authors state in their title that "trans-synaptic PTPd-IL1RAPL1 interaction regulates synapse formation", which is probably true but only inferred based on previous data and the assumed effect of the genetic tools used. It seems important to show at least that IL1RAPL1 is postsynaptic in the SLM (showing that the biochemical changes happen there will be more or less impossible).

→ This is an important question. However, unfortunately, we do not have antibodies that work for immunostaining of IL1RAPL1 proteins in hippocampal slices. We have to emphasize, however, that IL1RAPL1 is enriched in the PSD fraction relative to the P2 (crude synaptosomal) fraction (**Fig. 4B and G**), supporting its potential enrichment at postsynaptic sites in the SLM region, although our PSD samples were from the whole brain. In addition, previous studies strongly support the postsynaptic localization of IL1RAPL1. Specifically, IL1RAPL1 is enriched in PSDs and interacts with and promotes the synaptic localization PSD-95 (Pawlowsky, Gianfelice et al., 2010). HA-tagged IL1RAPL1 expressed in cultured hippocampal neurons is targeted to dendritic spines to attract excitatory presynaptic nerve terminals through trans-synaptic mechanisms and induce functional excitatory synapse formation (Valnegri, Montrasio et al., 2011, Yoshida, Yasumura et al., 2011). IL1RAPL1 deficiency in mice leads to a decrease in dendritic spine density in the hippocampus (Pawlowsky et al., 2010, Yasumura, Yoshida et al., 2014). We commented on these aspects with references in Results.

10. The behavioral data are intriguing but somewhat detached from the rest of the paper as Fig. 1-4 deal with regions of the brain that are probably not directly involved in the behavioral defects of the mutant mice. The discussion of these data reflects this detachment. I am not fully convinced that they need to be part of this study.

→ We agree with the reviewer's point. However, an important component of our study is to show the impacts of alternative splicing at both synaptic and behavioral levels, although these correlations should certainly be studied more in future studies. In addition, the face validity in the current study linking defective PTP δ with sleep dysfunctions would be important information to the researchers studying synaptic adhesion molecules (including PTP δ) and sleep disorders. We also discussed this issue with the editor. Lastly, although not requested, we added some more n numbers for the behaviors of *Ptprd-meA*^{-/-} mice (**Appendix Fig 1I and J and Fig EV4E and G**) to solidify our conclusion that *Ptprd*^{-/-} and *Ptprd-meA*^{-/-} mice show similar behaviors.

Conclusion:

I would really like to see this study published in EMBO Journal. But I think - particularly for EMBO Journal - a somewhat more elaborate analysis of the cell biology, which is at the core of the study, is warranted. In essence, even some of the

key conclusions are not watertight yet, e.g. the synapse-type specificity of PTPd. I think resolving the issues I outlined above is necessary, but I am willing to discuss this with the authors.

→ We believe that we tried our best to address all of the major comments raised. Again, we appreciate the very helpful comments of the reviewer.

Referee #2:

PTPs are thought to mediate adhesion between pre-and postsynaptic site but many details remain to be determined. Here Eunjoon Kim and co-workers show that PTPdelta has its highest expression level in the SML layer of the neuropil in the hippocampus with remarkably little of this protein present in neighboring layers including SR. Immuno-EM analysis indicates a mostly presynaptic localization of PTPdelta. KO of PTPdelta as well as elimination of the small meA insert that mediates binding to the postsynaptic adhesion protein IL1RAPL1 lead to reduction in the strength of synapses in SML but not SR or SO of the hippocampal CA1 area. In parallel, synapse number but not size is reduced in the SML of these mice. The authors identified IL1RAPL1 as the relevant interaction partner by an unbiased proteomic screen for changes in tyrosine phosphorylation, another strength of this manuscript. In this screen IL1RAPL1 showed by a large margin the highest change in tyrosine phosphorylation. According to these functional studies, PTPdelta and especially its bind to IL1RAPL1 has a selective role in mediating synapse formation in a define hippocampal layer.

Furthermore, full PTPdelta KO as well as conditional KO in glutamatergic but not so much GABAergic neurons and the meA deletion all result in hyperactivity as tested in open field and in Laboras cages. Finally, EEG and EMG recordings from the meA deleted and the conditional PTPdelta KO in glutamatergic neurons showed a reduction in nonREM sleep and delta power.

The experiments are very thorough and rigorously executed and conclusions as clearly supported and of high interest. I only have a modest concern

→ We appreciate the encouraging comments of the reviewer.

Modest Concern:

Fig. 2: that PTPdelta immunolabeling is largely not overlapping with MAP2B labeling is not truly indicative of the labeling being not within the dendrite. Microtubules and its associated MAP2B are in the center of dendrites and proteins in or near the dendritic plasma membrane typically do not overlap with MAP2B. However, this concern is alleviated by the immuno-EM analysis, which shows a mostly presynaptic localization of PTPdelta at least at the synapse.

→ We agree with the reviewer that the minimal colocalization of PTP δ -tdTomato signals with MAP2 does not necessarily mean that PTP δ -tdTomato signals are not in

dendrites. We clarified this point in both Results and figure legends.

Minor Concern:

The various CAM should be better defined. For instance, for somebody outside the field it is not clear what NGL3 stands. The authors might want to consider better defining all CAMs with full names in the introduction or provide a table with full names.

→ We provided full names for all the CAMs appearing in the Introduction.

Lastly, although not requested, we added some more n numbers for the behaviors of *Ptprd-meA*^{-/-} mice (**Appendix Fig 1I and J and Fig EV4E and G**) to solidify our conclusion that *Ptprd*^{-/-} and *Ptprd-meA*^{-/-} mice show similar behaviors. Again, we appreciate the very helpful comments of the reviewer.

Referee #3:

In this paper Park et al. studied extensively the function of the receptor tyrosine phosphatase PTPdelta. Using extensively mouse genetics they clearly showed that PTPdelta in mature neurons is essentially present only at the excitatory presynaptic membrane. The total deletion of PTPdelta in mice causes a specific decrease in excitatory synapse development and strength. Interestingly this is mediated by tyrosine dephosphorylation and synaptic loss of IL1RAPL1, a major postsynaptic partner. This was elegantly demonstrated by accurate phosphoproteomic analysis and using a PTPdelta-mutant mouse lacking the PTPdelta-meA insert, thus expressing a PTPdelta mutant lacking the ability to interact only with IL1RAPL1 among the other postsynaptic partners. Finally, they also provide strong evidence that mice deleted of PTPdelta exhibit abnormal sleep behavior and non-REM rhythms suggesting that specific alternative splicing in PTPdelta and trans-synaptic adhesion with IL1RAPL1 regulates excitatory synapse development and sleep behavior.

This is a very impressive and comprehensive paper that strongly demonstrates the synaptic function of PTPdelta. The data are very convincing and of top quality. The following two points should be addressed to clarify a couple of open questions.

→ We appreciate the encouraging comments of the reviewer!

1) The paper provides very nice and convincing data indicating that PTPdelta is a presynaptic protein localized at the excitatory synapses. They also showed, by EM analysis, that in the PTPdelta KO mice there is a reduction of PSD density in the SLM layer of the CA1, but no changes in the structure of the PSD. Considering that PTPdelta is a presynaptic phosphatase it should be important to analyze the structure of the presynapses by measuring for example size of the terminals and the density of synaptic vesicles.

→ In response, we performed the suggested EM quantification and found that the

density of presynaptic nerve terminals was decreased by ~28% in the *Ptprd*^{-/-} SLM, whereas the area of nerve terminals, or the density of presynaptic vesicles, was not decreased (**Fig 2P-R**). Considering the ~40% decrease in the density of PSD, these results suggest that presynaptic structures are substantially eliminated together with postsynaptic structures.

2) To further strengthen the finding that PTPdelta deletion specifically reduces the localization of IL1RAPL1 to PSD, it will be interesting to see if some of the other PTPdelta interacting postsynaptic CAMs remain localized at PSD in PTPdelta and PTPdelta-meA KO mice. If antibodies are available the authors should analyze the biochemical distribution on PSD fraction of some of the other postsynaptic CAMs interacting with PTPdelta.

→ We performed the requested experiment and found, in *Ptprd*^{-/-} mice, that other PTPδ-binning postsynaptic adhesion molecules such as Slitrk3, IL1RacP, and NGL-3 in synaptic fractions (P2, SPM, and PSD) were normal, being comparable to those in WT samples (**Fig EV3G**). Similarly, in *Ptprd-meA*^{-/-} mice, levels of Slitrk3, IL1RacP, and NGL-3 in synaptic fractions (P2, SPM, and PSD) were normal (**Fig EV3H**). These results contrast with the selective decrease in IL1RAPL1 levels.

Lastly, although not requested, we added some more n numbers for the behaviors of *Ptprd-meA*^{-/-} mice (**Appendix Fig 1I and J and Fig EV4E and G**) to solidify our conclusion that *Ptprd*^{-/-} and *Ptprd-meA*^{-/-} mice show similar behaviors. Again, we appreciate the very helpful comments of the reviewer.

References:

- Chagnon MJ, Uetani N, Tremblay ML (2004) Functional significance of the LAR receptor protein tyrosine phosphatase family in development and diseases. *Biochem Cell Biol* 82: 664-75
- Pavlovsky A, Gianfelice A, Pallotto M, Zanchi A, Vara H, Khelifaoui M, Valnegri P, Rezai X, Bassani S, Brambilla D, Kumpost J, Blahos J, Roux MJ, Humeau Y, Chelly J, Passafaro M, Giustetto M, Billuart P, Sala C (2010) A postsynaptic signaling pathway that may account for the cognitive defect due to IL1RAPL1 mutation. *Current biology* : CB 20: 103-15
- Valnegri P, Montrasio C, Brambilla D, Ko J, Passafaro M, Sala C (2011) The X-linked intellectual disability protein IL1RAPL1 regulates excitatory synapse formation by binding PTPdelta and RhoGAP2. *Human molecular genetics* 20: 4797-809
- Yasumura M, Yoshida T, Yamazaki M, Abe M, Natsume R, Kanno K, Uemura T, Takao K, Sakimura K, Kikusui T, Miyakawa T, Mishina M (2014) IL1RAPL1 knockout mice show spine density decrease, learning deficiency, hyperactivity and reduced anxiety-like behaviours. *Scientific reports* 4: 6613
- Yoshida T, Yasumura M, Uemura T, Lee SJ, Ra M, Taguchi R, Iwakura Y, Mishina M (2011) IL-1 receptor accessory protein-like 1 associated with mental retardation and autism mediates synapse formation by trans-synaptic interaction with protein tyrosine phosphatase delta. *The Journal of neuroscience : the official journal of the Society for Neuroscience* 31: 13485-99

2nd Editorial Decision

13 March 2020

Thank you for submitting your revised manuscript to The EMBO Journal.

Your study has now been seen by the two original referees and their comments are provided below. As you can see, the referees appreciate the introduced revisions and support publication here. I am therefore very pleased to accept the manuscript for publication here. Before I can send you the formal accept letter there are just a few remaining things we should sort out in a final revision.

- The reference format should be updated to fit the EMBO Journal style.
- You need to add the appendix figure plus legend to an appendix with a ToC. . The two tables uploaded as appendix tables should be re-labeled as Dataset EV1 and EV2 and their callouts need to be adjusted in the text.
- The proteomic data set should be deposited in a database and the accession number provided in the in the data availability section
- We encourage the publication of source data, particularly for electrophoretic gels and blots, with the aim of making primary data more accessible and transparent to the reader. It would be great if you could provide me with a PDF file per figure that contains the original, uncropped and unprocessed scans of all or key gels used in the figure? The PDF files should be labeled with the appropriate figure/panel number, and should have molecular weight markers; further annotation could be useful but is not essential. The PDF files will be published online with the article as supplementary "Source Data" files.
- I have asked our publisher to do their pre-publication checks on the paper. They will send me the file within the next few days. Please wait to upload the revised version until you have received their comments.

That should be all.

Congratulations on a nice study

REFEREE REPORTS

Referee #1:

The authors made a serious effort to deal with my comments. I think they did a very good job. I support publication.

Referee #3:

The authors fully addressed my requests.

2nd Revision - authors' response

17 March 2020

The authors performed all minor editorial changes.

Corresponding Author Name: Eunjoon Kim

Manuscript Number: EMBOJ-2019-104150